# communications
# engineering

## PERSPECTIVE

# Opportunities and challenges for sweat-based monitoring of metabolic syndrome via wearable technologies

Lynnette Lyzwinski[1,3], Mohamed Elgendi [2,3✉], Alexander V. Shokurov[2],
Tyler J. Cuthbert[2], Chakaveh Ahmadizadeh [2] & Carlo Menon [1,2✉]

Metabolic syndrome is a prevalent condition in adults over the age of 65 and is a risk factor for developing cardiovascular disease and type II diabetes. Thus, methods to track the condition, prevent complications and assess symptoms and risk factors are needed. Here we discuss sweat-based wearable technologies as a potential monitoring tool for patients with metabolic syndrome. We describe several key symptoms that can be evaluated that could employ sweat patches to assess inflammatory markers, glucose, sodium, and cortisol. We then discuss the challenges with material property, sensor integration, and sensor placement and provide feasible solutions to optimize them. Together with a list of recommendations, we propose a pathway toward successfully developing and implementing reliable sweat-based technologies to monitor metabolic syndrome.

In recent years, metabolic syndrome has been increasingly recognized as a public health problem[1]. It is a condition that predisposes individuals to an increased risk of developing type II diabetes[2], cancer[3], lung function impairment[4], and cardiovascular disease[5]. The reported prevalence of metabolic syndrome is high, approximately 30–40% of adults over the age of 65 suffer from it[6]. A recent systematic review found that 23.7% of adults worldwide living with type I diabetes mellitus have metabolic syndrome[7]. Similarly, high estimates were found among obese adults in Europe, with 24–65% of obese women having metabolic syndrome and 43–78% of obese men[8]. There has been a steady rise in the global prevalence of the syndrome over the last decades[9], and the trends have shown a higher prevalence in older age groups (e.g., over age 70 years)[10]. These estimates are also contingent upon the definition of metabolic syndrome and there are varying definitions summarized in Table 1. Typically, patients present with a classic triad of signs: obesity, hypertension, and high fasting glucose levels[11]. Other signs include high levels of low-density lipoprotein (LDL), low levels of high-density lipoprotein (HDL), and wide waist circumference, though only three of the above clinical parameters are needed for a diagnosis[11]. According to the current guidelines of the National Heart, Lung, and Blood Institute, patients with fasting blood glucose levels of 100 mg/dL and above, HDL levels below 50 mg/dL (40 mg/dL for men), triglycerides of 150 mg/dL and over, blood pressure greater than 130/85 mmHg, and a waist circumference of over 35 inches (40 inches for men) have a substantially increased risk of cardio-metabolic complications[12].

Individuals with metabolic syndrome who also have a high risk of complications have an approximately 20% greater chance of developing a myocardial infarction over 10 years[12]. Furthermore, a longitudinal study found that glucose [insulin intolerance incidence rate ratio (IRR) = 1.81], overall body weight, and blood lipid levels are predictors of type II diabetes

[1] Menrva Research Group, Schools of Mechatronic Systems Engineering and Engineering Science, Simon Fraser University, Metro Vancouver, BC, Canada.
[2] Biomedical and Mobile Health Technology Lab, Department of Health Sciences and Technology, ETH Zurich, Zurich, Switzerland. [3] These authors contributed equally: Lynnette Lyzwinski, Mohamed Elgendi. ✉email: moe.elgendi@hest.ethz.ch; carlo.menon@hest.ethz.ch

**Table 1 Different clinical definitions of metabolic syndrome[18,167-173].**

| Organization | Year | Obesity | Hyperglycemia | Dyslipidemia | Hypertension | Additional health condition | Diagnostic criteria | Applicability to Sweat-Based Wearables |
|---|---|---|---|---|---|---|---|---|
| World Health Organization (WHO)[168,173] | 1999 (original), 2004 | Waist-to-hip ratio (0.9 men; 0.85 women) | Glucose intolerance/diabetes/insulin resistance | Triglycerides >150 mg/dL; HDL ≤35 mg/dL men; HDL ≤39 mg/dL women) | >140/90 mmHg | Microalbuminuria 30 ugm/mg; Albumin/creatine ratio | One of the glucose intolerance issues described plus two others from the list | +++ |
| National Institutes of Health[170] | 2001 | Abdominal obesity (WC = 102 cm men; WC = 89 cm women) | High blood glucose >100 mg/dL | Triglycerides >150 mg/dL; HDL <40 mg/dL | >130/85 mmHg | | At least any three | +++ |
| National Cholesterol Education program (NCEP)[71] | 2003 | WC ≥102 cm men; WC ≥88 cm women | Fasting blood glucose >110 mg/dL | Triglycerides >150 mg/dL HDL <50 mg/dL women; HDL <40 mg/dL men | >130/85 mmHg | | Any three of more of | +++ |
| American Association of Clinical Endocrinologists[167] | 2003 | Obesity (BMI 25 kg/m², especially higher cutoffs >35 kg/m²) Waist-to-hip ratio = 0.91 | Insulin intolerance/hyperinsulinemia | Triglycerides >150 mm/HG; Low HDL | Elevated blood pressure; Hypertensive (>130/85 mmHg); Sodium retention | Plasma and urine uric acid elevation; Elevated fibrinogen; Prothrombotic state; Inflammatory markers; C-reactive protein elevation; Endothelial dysfunction | Three or more in addition to the clinical history of the patient | ++ |
| International Diabetes Federation (IDF)[172] | 2006 | Abdominal obesity/Elevated WC (according to race; Europeans: WC ≥94 cm men, WC ≥80 cm women; Africans: same as the above; South Asians WC ≥90 cm men, WC ≥80 cm women; Chinese and Japanese: | Fasting blood glucose >100 mg/dL | Triglycerides > 150 mg/dL; HDL <50 mg/dL women; HDL <40 mg/dL men | >130/85 mmHg | | Abdominal obesity plus two of the other criteria listed here | +++ |

**Table 1 (continued)**

| Organization | Year | Obesity | Hyperglycemia | Dyslipidemia | Hypertension | Additional health condition | Diagnostic criteria | Applicability to Sweat-Based Wearables |
|---|---|---|---|---|---|---|---|---|
| National Heart, Lung, and Blood Institute | 2022 Updated | WC ≥ 90 cm men, WC ≥ 80 cm women) WC > 40 inches men WC > 35 inches women | Fasting glucose ≥100 mg/dL | Triglycerides > 150 mg/dL; HDL < 50 mg/dL women; HDL < 40 mg/dL men | >130/85 mmHg | | Three or more | ++++ |

+ = Low applicability, ++ = Moderate applicability, +++ = High applicability, ++++ = Very high applicability. If a waist-to-hip ratio is required, it is more complex for sensors than a simple waist circumference (WC) measure. More clinical indicators that can be measured in sweat, such as uric acid, C-reactive protein = higher applicability.

development over a 4-year period, with a 37–52% higher incidence risk than their counterparts[13]. The risk of diabetes for patients with metabolic syndrome is reduced by lifestyle modification, which involves dietary changes[14]. High sugar intake and sugar-sweetened beverages have been linked to an increased risk of metabolic syndrome and associated complications. Individuals who consume high amounts of sugar are 32% more likely to develop metabolic syndrome than their counterparts[15].

Moreover, research indicates that patients with metabolic syndrome are more likely to have sodium sensitivity, whereby their bodies may have a greater response to sodium. Subsequently, they may have significantly elevated blood pressure after sodium intake when compared to individuals without metabolic syndrome[16]. Further research has found that sodium intake (levels excreted via urine) has a positive association with several key elements of metabolic syndrome besides hypertension, which includes increased body fat, increased weight, insulin resistance, and inverse relationships with protective HDL[17]. Specifically, individuals who consumed the highest amounts of sodium were 1.92 times likelier to report having metabolic syndrome when compared to their counterparts who consumed lower levels of sodium (95% CI: 1.6–2.2; $p < 0.01$)[17].

A proinflammatory state, recognized clinically by elevations of C-reactive protein (CRP), is commonly present in persons with metabolic syndrome[18,19]. Research in men has found that concentrations above 3 mg/dl significantly increase the risk of Metabolic syndrome by 3 fold relative to men with less than 1 mg/dl of CRP[19]. Specifically, CRP has been linked to the development of hypertension in individuals who previously had normal blood pressure, and concentrations above 3 mg/dl increase the risk of cardiovascular disease (CVD)[20–23]. High CRP levels have also been found to be associated with an increased risk of developing insulin resistance and type II diabetes[19,24]. Additionally, patients with existing metabolic syndrome are at a much higher risk of developing cardiovascular disease when high sensitive C-reactive protein levels are elevated >3.0 mg/l[19].

Thus, there is a need to develop effective and scalable interventions for metabolic syndrome[6,11]. In recent years, mobile health (mHealth) technology has emerged as a medium for promoting behavior changes and reducing lifestyle-related risk factors associated with chronic conditions, such as obesity[25]. mHealth technology also has the potential to assess metabolic syndrome and its associated risk factors and clinically relevant parameters via the use of emerging wearable smart clothing—textiles, wristbands, rings[26] (e.g., Oura ring)[27], and smartwatches—that collect biomedical data from subjects with different health conditions[28–30]. Textile-based devices (i.e., clothing) have the potential to decrease the barrier of access to biosignal monitoring since they are familiar to essentially all populations, sit on/close to the body, and can be employed on all parts of the body, which may increase acceptance, enable capture of different signal types, and expand accessible areas of the body for monitoring, respectively. In addition, textile-based sensors should ideally possess similar characteristics and usability to traditional textiles/clothing to allow ease of use and direct integration into daily lives, such as ensuring biocompatibility to allow long-term use (and reduce the chance of irritation often caused by medical devices) and compatibility with common cleaning/washing machines to ensure devices can be (re)used daily. By gaining insight into vital health parameters, patients with metabolic syndrome and their physicians can tailor their treatment plans accordingly and improve their overall assessment of patient risk.

One novel method for detecting signs and clinical parameters associated with the syndrome involves the use of sweat-based technology. Sweat contains various health-related biomarkers, including ascorbic acid, uric acid, metabolites, electrolytes, small

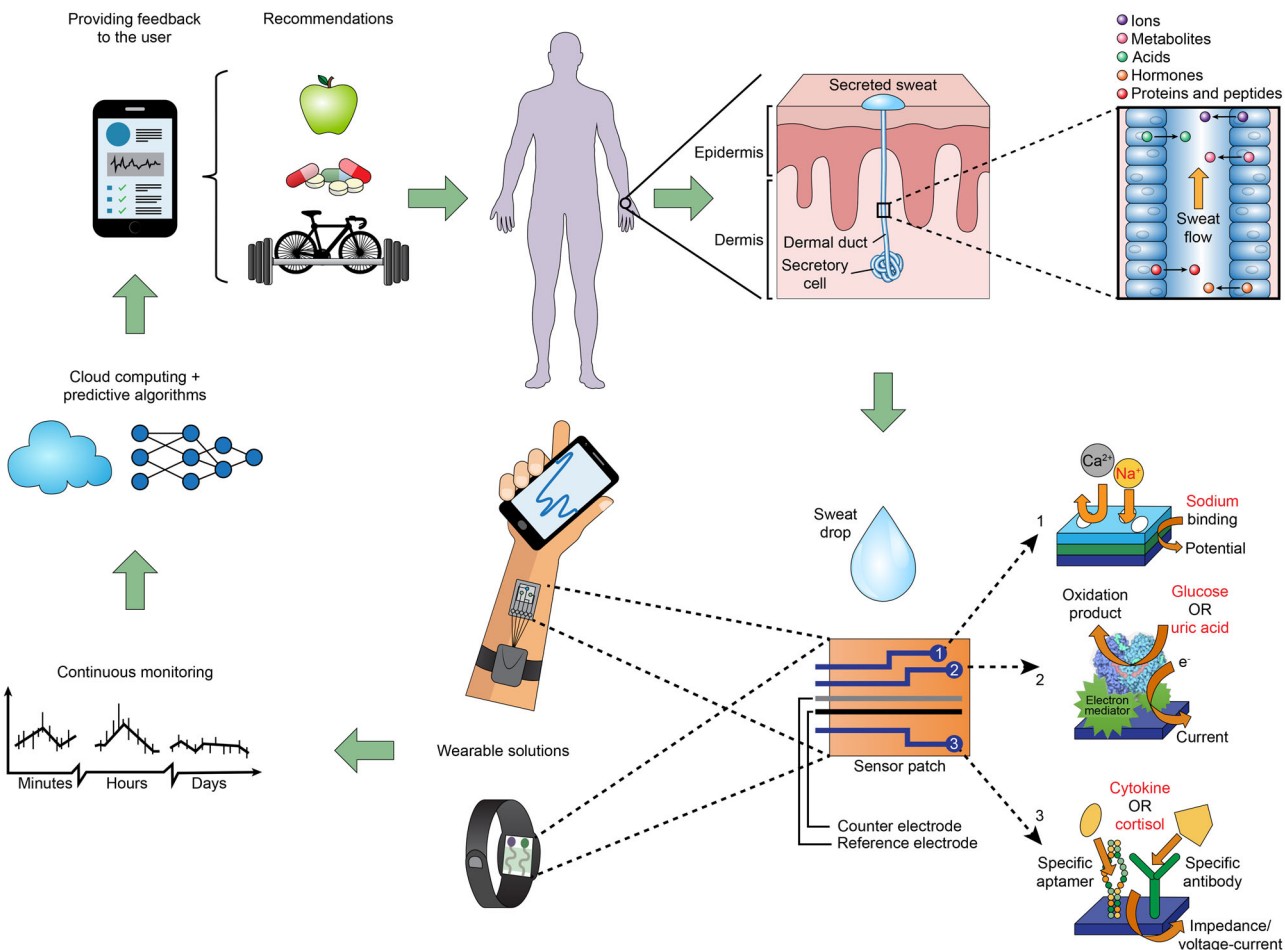

**Fig. 1 Wearable sensor patch integrating multiple technologies for sweat biomarker analysis.** The figure highlights the full cycle, starting with sweat production and ending with personalized feedback for the user. (1) Solid-contact ion-selective electrodes for $Na^+$, which can be based on an ion-selective membrane that utilizes ionophores—molecules capable of encapsulating a specific ion within a membrane—and a transduction layer responsible for the generation of analytical signals in the form of electrochemical potential. (2) Enzyme-based electrochemical sensors with immobilized enzymes that can selectively oxidize glucose or uric acid (creating glucose oxidase and uricase, respectively). Combined with the proper materials, these enzymes can be transduced into an electric current, which can be translated into precise biomarker concentrations. (3) Specific binding agent biosensors with immobilized aptamers or antibodies for the detection of cortisol and cytokines. Both aptamers and antibodies allow for the specific recognition of small molecules and proteins; the act of biomarker binding changes the electrochemical properties of the electrode surface, which can be treated as an analytic signal. This conceptual patch also contains reference and counter electrodes, usually fabricated using inexpensive inorganic materials, which are required for the realization of the electrochemical measurements used in (1)–(3).

proteins, and steroid hormones that may be passively collected and analyzed[31,32]. The following clinical indicators of the metabolic syndrome may be assessed via sweat according to the various definitions of the syndrome summarized in Table 1.

Most state-of-the-art developments regarding wearable real-time biomarker detection rely on electrochemical techniques that are robust, have low detection limits, and can be easily miniaturized and integrated into electronic components[33]. Although biomarkers represent different chemical classes and require different analytical techniques, there is potential for a consolidated sensor. Figure 1 illustrates the full cycle of sweat production and monitoring in a proposed futuristic technology. The figure starts by showing how sweat is produced by sweat glands and travels to the surface of the skin. The proposed technology involves a wearable patch that combines three different technologies to detect five metabolic biomarkers in the sweat. These biomarkers include glucose, lactate, sodium, potassium, and cortisol, which are continuously monitored and processed on the cloud via artificial intelligence. The data is then used to provide personalized feedback and recommendations to the user, such as advice

on healthy eating or physical activity. This figure provides a comprehensive overview of the proposed technology and highlights the potential of sweat-based wearable technology for monitoring metabolic health in a non-invasive and personalized way.

Analysis of cationic concentrations (e.g., $Na^+$ and $K^+$) can be performed using highly sensitive, solid ion-selective electrode technology[34]. Glucose-sensing can be performed using enzymatic or non-enzymatic oxidation, which is then transduced into electric signals on an electrode[35]. The presence of small, non-charged molecules, such as cortisol, can be sensed with electrodes composed of molecularly imprinted polymers[36]. Specific antibodies and metabolic aptamers can be used to reliably recognize both small molecules (such as cortisol) and biomarker proteins (such as cytokines)[35,37]. Combining these sensing technologies into a single wearable device that has the potential to accurately identify and monitor metabolic syndrome could have a huge impact on public health. By providing a clear clinical profile of patients and their behavioral risk factors, complications associated with metabolic syndrome would decrease and, thus, health

outcomes would be improved. In the following sections, we review the state of the art in sweat-based wearable sensing devices and discuss the challenges towards achieving our vision of an integrated device for clinical diagnosis.

## Targets for sweat-based metabolic syndrome assessment

A future integrated sensor for metabolic syndrome should ideally measure as many parameters as possible to obtain a comprehensive assessment of an individual's metabolic health. However, for the purpose of this study, we focused on five specific analytes (glucose, sodium, CRP, uric acid, and cortisol) that have been identified as relevant for the diagnosis and management of metabolic syndrome according to the 2022 NIH definition provided in Table 1. It is worth noting that the optimal sensor for metabolic syndrome may vary depending on the specific diagnostic criteria and patient population being studied. These five targets satisfy the most updated definition (as well as past definitions) of metabolic syndrome assessment published by the National Institutes of Health (NIH) in 2022:

1. Glucose, the primary metabolite for energy production in the body[38], is elevated in patients with metabolic syndrome[39]. Sweat glucose will be used to monitor glucose levels in the body in this perspective.
2. Sodium, an electrolyte[40], is associated with elevated blood pressure[41].
3. Cytokines (CRP) indicate inflammation in the body and are elevated in metabolic syndrome patients with risk factors for cardiovascular disease[42].
4. Uric acid levels are a risk factor for insulin resistance[43] and indicate hypertriglyceridemia (when serum triglycerides elevate in the presence of hyperuricemia and rise with increasing uric acid levels)[44]. They are also linked with obesity[45] and are risk factors for abdominal obesity[46].
5. Cortisol levels are linked to and are risk factors for abdominal obesity[47].

The applicability of the sweat-based wearables in monitoring these biomarkers ranges from moderate to high (see Table 1). For example, the 2022 NIH definition is ranked with the highest applicability to wearables, as it requires the presence of any three out of four indicators (obesity, hyperglycemia, dyslipidemia, and hypertension), while the 2003 American Association of Clinical Endocrinologists definition is raked moderate in terms of wearables applicability due to the requirement of collecting specific targets in addition to accessing the patient's clinical history.

**Measuring glucose levels with sweat-based wearables**. We found that glucose was commonly measured in sweat via wearables[48–59]. These included wearables with sweat patches or absorbent wrist bands. For example, He et al. developed a textile-based sensor (fabric with strong electrical conductivity made of silk) that collected data on glucose levels excreted via sweat that could be easily worn as a patch[50]. The Wearable Awareness Through Continuous Hidrosis sensor for sweat-based glucose monitoring was validated; it had a high correlation with standard glucose measuring methods, and the Bland-Altman plot demonstrated good comparability[54]. Researchers recently developed a comprehensive epidermal sweat patch with ultrasound transducers and found that glucose levels measured by sweat correlated with eating and exercising in participants[55]. Lee et al. developed a wearable enzyme-based, glucose-sensing patch (or disposable strip) that also adjusted glucose levels in patients transdermally via nanoparticles by using hyaluronic acid as a base (this patch utilized a micro-needling technique)[60]. Most of the studies we included were performed on a small number of healthy subjects,

and many studies did not validate their findings (i.e., by comparing their sweat sensors to standard capillary blood glucose measurements).

It should be noted that glucose levels in sweat correlate to blood glucose concentration[61], which is considered a diagnostic value for diabetes and metabolic syndrome. Despite that, measuring glucose in sweat remains a highly challenging task. While blood glucose levels do indeed correlate with sweat glucose, this correlation can differ in patients[62], which makes constant calibration necessary. And while some very recent advances[63] enable self-calibrating sensors to be produced; they are yet to be validated in clinical trials. Real-time analysis of glucose in sweat is also troubled by factors inherent to the very medium being analyzed and its natural environment. Sensor readings can be severely affected by contamination of the skin where sweat is being collected and analyzed, and the change of contact between skin and the sensitive elements because of epidermis flaking[64].

A recent study[65] looked into the composition and mechanisms of contaminants in sweat, which can accumulate on the surface of sweat-based sensors and interfere with their accuracy in measuring mineral levels. These contaminants can include proteins and lipids, which can result in inaccurate measurements and reduce the reliability of the sensor. This buildup of contaminants on the sensor's surface is referred to as "biofouling,"[66] and it can occur over time due to various factors, such as the composition of the sweat, the design of the sensor, and the duration of use. Although cleaning the skin before measuring can help in a clinical setting, it is not practical to do so regularly during mobile sensor use. To address this issue, researchers[67] are exploring alternative methods such as selective membranes, anti-fouling layers, and enzyme stabilizers, which can prevent contaminants from sticking to the sensor surface and improve the sensor's accuracy and reliability.

The sensing modalities commonly used in wearable devices that can be miniaturized rely on enzymatic reactions, which offer high selectivity and efficient analytical performance. However, enzyme-based sensors are known to have limited shelf-life, as demonstrated by several studies[68,69], which highlights the need for further research in this area, particularly for wearable glucose sensors. To address this issue, current research is focused on exploring non-enzymatic pathways for glucose detection, which offer the promise of extended shelf-lives[70].

The stability and reusability of materials used in sweat glucose sensing in non-enzymatic variety require substantial improvements before widespread use[71]. As such, sweat biofouling of sensors that require intimate contact with skin remains an unanswered challenge[65]. At the same time, some studies considered in the present perspective have demonstrated state-of-the-art materials capable of lasting for weeks or months. Addressing chemical-analysis-related challenges, such as the structure of matter in time and space needs fast, automated, stable, accurate, sensitive, selective, and even in situ analytical methods and protocols, would require multidisciplinary effort to provide a reliable wearable solution tested and evaluated in clinical studies with a large number of participants.

**Measuring sodium with sweat-based wearables**. We identified several studies that measured sodium levels in sweat, mostly conducted on small samples of healthy athletes[49,50,72–77]. Gao et al. developed a new plastic biosensor with silicon circuits that allowed for optimal electrical conductance and continuous 24-h monitoring and analyzation of multiple sweat components, including sodium[77]. Therse-Thakoor et al. developed a textile-based fabric patch that was ion and pH sensitive; when on the

skin's surface, it continuously monitored sweat excretion, the data of which was transferred wirelessly to a smartphone[76].

Most of the studies demonstrated good stability during exercise, as the various sensors continuously monitored the composition of sweat during changing physiological conditions[49,72–76]. Indirect validation was performed by a few studies that examined changes in sweat under altered physiological conditions (e.g., changes in electrolyte balance) during exercise[49,72,73,75,76]. A study investigating sodium levels in patients with clinical metabolic syndrome using wearable biosensors has not been undertaken to date. While recent studies show that sweat sodium concentration is almost certainly independent of its blood concentration[78], this value for sweat is now considered a good diagnostic value by itself, along with sweat chloride levels[79]. Thus, adding sodium sensors to textile-based clothing as part of an integrated approach to digital metabolic syndrome monitoring and wellness (which includes glucose tracking in real time, along with other parameters) may be valuable in future research and development.

**Measuring inflammatory biomarkers with sweat-based wearables**. Emerging wearable technology research suggests that CRP levels, along with other inflammatory markers (interleukins), may be measured using sweat-sensing wearable patches[80]. A proof-of-concept study was performed on a clinical population of patients with inflammatory bowel disease (IBD). The researchers found that the patches could accurately detect CRP levels in the participants' sweat. When they compared their measurement method against a reference enzyme-linked immunosorbent assay (ELISA), they found high levels of agreement and correlation[80]. Several studies on relationships between levels of inflammatory proteins in sweat and blood have shown correlations[81], further supporting the feasibility of measurement of these biomarkers in sweat for diagnostic applications.

**Measuring uric acid levels with sweat-based wearables**. We identified a few studies that measured uric acid levels in sweat[50,82–84]. He et al. made a comprehensive silk textile-based carbon sweat patch which not only measures glucose and sodium, but also uric acid levels. They found that the patch had good stability for monitoring uric acid levels in real time[50]. Xu et al. used an electrochemical flexible sensor with hydrogel to measure uric acid levels excreted via sweat, also finding that it had good stability for continuous monitoring under different physiological conditions[84]. Another study measured uric acid levels with a sensor that contained nickel and zinc hydroxide and found good re-test repeat reliability[82]. More research is needed to compare uric acid levels in sweat with blood and urine levels to validate these findings.

**Measuring cortisol with sweat-based wearables**. A future wearable for monitoring metabolic syndrome could also collect data on cortisol levels to mitigate the risk in patients. Researchers have found that patients with metabolic syndrome have higher amounts of an enzyme (11B-HSD1) that converts the inactive stress hormone cortisone to cortisol in the liver[85,86], and high cortisol levels are associated with an increased risk of premature mortality from CVD. It has been previously theorized that signs associated with metabolic syndrome—including adiposity, high blood pressure, and insulin resistance—could be reduced if cortisol levels are controlled[87]. Research indicates that patients with high cortisol levels (measured via urinary excretion) were five times more likely to die from CVD over a 6-year period[88] than their counterparts with lower cortisol levels.

We also identified one study that measured both glucose and cortisol as metabolic indicators[54]. Each participant wore a watch that passively collected sweat (labeled as a natural sample). The study was validated by both a high correlation ($r = 0.86$), and a mean absolute relative difference (MARD) of only 5%, between the cortisol measured by the watch and the gold standard of cortisol measurement[54]. The use of sweat for cortisol measurement was similarly validated in a study (employing wearable wristband sensors) that used chronoamperometry via a polarization technique on a circuit through which a current traveled[71]. It is important to underline that sweat cortisol levels were previously found to be correlated to serum levels[89], making non-invasive measurements by wearable devices a promising tool for diagnosing metabolic syndrome.

**Analysis**. We aimed to discuss the different types of sweat-based technologies that have been employed to assess one or more relevant clinical parameters to Metabolic syndrome. We identified proof-of-concept studies that have been undertaken to monitor glucose, sodium, and CRP levels in this review. However, there is a need for actual clinical trials in larger populations, instead of sampling the technology on a small number of healthy participants. We did not find any studies that evaluated smart wearables or biosensors for dyslipidemia, including those that collected blood and measured cholesterol (HDL and LDL) and other lipoprotein levels. Nor did we identify any wearable technology that identified biomarkers relevant to a prothrombotic state, including data on fibrin and fibrinogen levels. Finally, although abdominal obesity is a clinical indicator of metabolic syndrome, we did not identify any smart wearable devices that assess it.

Most of the studies demonstrated good repeatability making the sensors reliable for providing benchmarked results[90]. However, validity tests (i.e., checking for the accuracy of the measures against the gold standard traditional measures) were less frequently performed. Additionally, MARD, Clark Error Grids, and Bland-Altman plots were reported in only a handful of studies[54,55,57,80,91–94], and most researchers conducted correlation analyses by comparing their sensor readings against the results of the gold standard methods of measurement. Most of the studies were undertaken in healthy adults and tested for proof of concept with a small group of individuals, highlighting the gap in research concerning validation in large clinical studies.

**Other biomarkers**. Biosensors that collect data on fibrinogen and cholesterol would complete a comprehensive mobile wearable that will provide the best clinical diagnostics. Fibrinogen is a marker of inflammation in the body and a risk factor for blood clots[95,96]. Patients with elevated fibrinogen levels are at risk of a cardiovascular event, including premature mortality from a myocardial infarction[97], and also elevated in Metabolic syndrome patients[98–100]. Some medications can lower fibrinogen levels in the blood[98]. By contrast, a healthy diet rich in fruit and vegetables, such as the Mediterranean diet, is linked with lower fibrinogen[101]. Data on cholesterol is also essential because it is elevated in patients with Metabolic syndrome[102] and puts them at risk for a cardiovascular event. Again, it may be modified with diet and medication[103,104].

While detection and quantification of these biomarkers in sweat would be of increasing use to real-time diagnostics using wearable devices, there is still much to be known about them. Evidence on correlations between sweat and blood cholesterol levels and fibrinogen is lacking. With the lack of this knowledge, developing wearable sensors for these analytes remains in its infancy. The effort from clinical scientists to elucidate possible relationships and materials researchers to figure out possible

sensing modalities are needed. Materials science will follow if diagnostic correlations for concentrations of these molecules in sweat can be found for metabolic syndrome or other conditions. It is known that modern advances in antibody-based assays and molecularly imprinted polymer technology can produce sensitive elements for an ever-increasing list of analytes[105].

## Device design

**Sensing materials**. Because metabolic syndrome has multiple clinical parameters, it is doubtful that a single sensor can diagnose and monitor its risk factors. When developing a sensor for each parameter, certain factors must be considered from an engineering design standpoint, such as the optimal sensing material for monitoring metabolites, the optimal anatomical site for monitoring these metabolites, and the stability of the sensors and their robustness.

A previous review weighed the pros and cons of various materials for biosensors[106]. They considered the use of nanoporous gold-plated materials, given their flexibility. However they argue that the costs of using gold make using this material less feasible and practical[106]. Additionally, graphene was considered a good material for biosensors, including multiplexed ones, given its thermal and electrical conductivity properties. Still, limitations include its ability to operate in a range of external environmental conditions[106]. They also consider carbon nanotubes, as they have sufficient thermal capacity, conductance, durability, and stability[106]. However, their drawbacks include toxicity; even when coated with chitosan, adding metallic nanotubes is toxic and costly[106]. Mesoporous carbon has all of the previously mentioned benefits above but has the additional benefit of flexibility and porous structure, but drawbacks include the high costs of associated regents and toxicity[106]. The use of nanoparticles has also been previously recommended to increase the sensitivity of glucose detection, particularly during unstable conditions such as food packaging in the case of food glucose sensors[107]. This, however, may be considered when enhancing the sensitivity of human sweat glucose detection under less stable lab-controlled conditions (e.g., changes in sweat excretion rate under different physiological conditions and contamination via water during showering, for example).

The most common sensors used in the studies we analyzed were patches[48,50,55,72–74,76,80,84,108–110], bands/wristbands[51,59,73], and watches[49,54,58]. In terms of optimal sensing material, a variety of different materials was used in the literature. The studies that validated their findings against the gold standard measures used flexible carbon-based materials[50,83,111], zinc oxide electrodes with nanopores[58,91,92], and electrochemical nanocomposite materials[55,57,83,92]. Many studies (42.8%) explicitly described the flexible or stretchable fabric used for the sensors[50,55,57–59,72,74,76,77,83,84,91,92,110,112,113]. In several studies, thread-based fibers were used[48,59,76,114], specifically, carbon-based fibers[50,83,111] and those with nanocomposites or nanoparticles[55,57,83,92]. Some of the materials (14%) had an ion selective membrane that allowed for electrolyte detection via potentiometric sensor-based fabric[72–76]. A few studies (8.5%) used zinc oxide[58,91,92]. Other studies used polymeric materials, such as polyethylene[77,115] and sodium polyacrylate[109], or gel-based materials (14%)[84,108–110], including siloxane-based components[52,53], for the sensors. A few studies used 3D printing[48,75,113,114]. Microfluidic materials were also used in a few studies (25%)[48,49,59,72,73,75,84,108,116].

Several studies used enzymatic reactions[72,76,84,91,92] to measure the biomarkers' percentage levels, but most used electrode-based sensors involving electrochemical reactions (48.5%)[49–51,55,57,58,72,82–84,91,92,108–110,112,115]. Some studies'

sensors were disposable (14%)[51,82,109,110,115]. A few studies (11%) used smartphones that paired with the sensors to assist with the readings of the measured biomarkers[48,59,76,109].

Also, sensors capable of powering themselves via (electro-) chemical reactions are promising advancements for wearable sensors to eliminate the dependence on batteries[117]. However, there are still plenty of challenges in advancing these concepts into practical devices. A key component of these devices that may require consideration is the biocompatibility of the novel components and materials with the skin since they are often unknown during the early stages of development. Materials[118,119] that combine a high degree of selectiveness and specificity, necessary for ion-sensing, glucose, uric acid, as well as cortisol/cytokine sensing, with anti-fouling and self-cleaning properties, in a user-friendly form such as textiles, would greatly enhance widespread sensing. This advancement could contribute to improving our understanding of the health status of a larger segment of the general population. However, much research is still required to achieve such materials.

**Sensor stability**. Many studies explicitly indicated that the sensors demonstrated good stability under different physiological conditions over long periods of time[57–59,72–75,80,82,91,110,113–115], and as most of the tests were done during periods of exercise with varying body positions, the sensors also proved robust. The ideal sensor must be stable and durable to withstand continuous wear and changing environmental conditions such as exposure to water. Ageing, or degradation over time in storage or use, of biological products such as enzymes and antibodies used in biosensors can lead to decreased sensitivity and low reliability[120]. Although biosensor ageing is a well-studied issue, stabilizing biological components in sensors remains a significant challenge for successful implementation in devices[121]. Not only biological, but also polymeric and inorganic components of sensors can degrade over time in storage and/or use[120]. Therefore, the ageing of materials must be considered, especially for textile wearable sensors that are subject to more mechanical stress during usage and washing[122]. Signals can no longer be transmitted efficiently when materials degrade, which can impact the reliability of the sensor[123].

Patients should be aware of the shelf life of the sensor and have replacement sensors after a certain period of use (e.g., washing, heat, wear and tear). For instance, durability via reduced permeability in polymer-based sensors that block water is a desirable feature of sweat-based sensors[124]. Some sensors may also be sensitive to changes in heat and PH[125,126]. Thus, in non-perfectly controlled conditions in the lab, extreme changes in heat and PH during exercise may influence readings if a sensor cannot withstand these conditions.

**Integrated wearable sensor**. The proposed technologies for use in a future multiplexed sensor are becoming well-established, capable of detecting specific biomarkers in sweat with robustness. However, further research is necessary to translate these foundational technologies into a comprehensive multiplexed sensing device. One challenge is the rapid biofouling that occurs in sweat, particularly in techniques that depend on molecular analyte-receptor interactions at the interface, such as molecularly imprinted polymers and aptasensors[127]. Additionally, contamination from sweat and skin can have a significant impact on sensor performance, as discussed in the section on enzymatic sensors. Advanced materials are being developed to address these challenges by incorporating anti-fouling properties and self-cleaning capabilities[128]. This ongoing research aims to improve the robustness and reliability of future multiplexed sensors.

Recent reviews of multisensory systems have found that they enhance the overall predictive ability to detect relevant health events, acting in unison, with greater reliability in their readings[129,130]. However, the integration of multiple sensing modalities may result in electronics crosstalk, which can be resolved with appropriate data acquisition electronics. However, chemical crosstalk can still be a potential issue when multiplexing biosensors on the same chip, particularly in the case of enzymatic sensors that generate hydrogen peroxide upon interaction with the analyte. Recent studies[131,132] have proposed microfluidic systems as a solution to this problem, where sweat flow can be split into separate channels corresponding to each individual sensor to prevent any chemical crosstalk and mixing of samples. This approach allows different sensors to analyze the same sweat sample simultaneously and in real-time. Such a strategy can be used to multiplex a wide range of sensors on a single microfluidic device[133].

Comfort also needs to be considered as not all patients may be comfortable with wearing multiple sensors that collect different measures[129]. The future of wearable sensors should involve the development of an integrated wearable piece of smart clothing that can collect data on multiple relevant clinical parameters and risk factors. Ideally, smart clothing with integrated sensors should be made of comfortable fabric that can be easily worn and allow for data from the Metabolic syndrome patient to be passively collected.

A future wearable should also measure abdominal obesity via waist circumference as these indicators are highly relevant to the syndrome, highlighted in Table 1. In addition, a future sensor for Metabolic syndrome could also measure physical activity. Specifically, low levels of physical activity may result in obesity and being overweight[134], high blood pressure[135], and insulin intolerance[136], all of which are associated with metabolic syndrome. The risk of Metabolic syndrome is reduced with increasing levels of physical activity and reduced sedentary behavior[137,138].

The syndrome could be monitored, and its behavioral risk factors could be mitigated when paired with a smartphone application that alerts when unhealthy behaviors are detected. Physicians could access their patients' data on demand and/or offer rapid care (via telehealth)[139] to remote patients without the need to invasively collect blood samples to test glucose and inflammatory marker levels. Patients could also be prioritized in terms of appointments with cardiology or endocrinology specialists. Metabolic risk profiles could be created for patients, and tailored advice would be provided to assist them with adjusting modifiable risk factors. Data from the sensors could be integrated into smartphone applications. Modeling risk over time, according to metabolic syndrome patient risk profiles, would be possible through continuous data collection. Machine learning has been previously used to compute the risk of cardiac events in patients based on their risk factors, including the likelihood of having a future myocardial infarction[140,141]. Thus, depending on a patient's risk for developing diabetes or cardiovascular disease, more intensive treatment and personally tailored advice could be given by using a machine learning algorithm.

## Sweat collection methods for monitoring metabolic syndrome
**Optimal anatomical site**. With regard to the anatomical site for monitoring biomarkers, most studies used the wrist[51,54,58,77,93] or arm[49,50,53,55,73,76,80,83,84,116] as the main sites for biomarker collection. However, a few studies demonstrated that sweat-based sodium may be collected on the forehead or leg. Two studies explicitly mentioned using the lower back[51,72]. Other body parts involved in data collection included the chest, leg, abdomen[108], and forehead[51].

A complex question that remains unanswered is where the optimal anatomical location for sweat analysis is. The variability and rate of sweat distribution over the surface of the human body can cause problems for the inter-participant accuracy of wearable sweat-analyzing devices[142]. For example, eccrine glands are present from birth and release secretions in the form of aqueous fluid that contains relatively more waterborne biomarkers, such as $K^+$ and $Na^+$[7]. Apocrine and apoeccrine glands, however, not only secrete more viscous fluid that contains more organic compounds, but also develop only after puberty and are localized around hair[143,144].

Studies[145,146] have shown that sweat rates vary significantly across the body, as evidenced by thermal mapping and sweat collection from specific areas of the body, as shown in Fig. 2a, b. A high sweat rate could potentially lead to dilution effects, which could negatively impact the accuracy of metabolite analysis for low-concentration metabolites. However, some low-concentration metabolites may not be present enough in the sweat to be accurately detected and quantified, particularly if the sweat rate is too low[35]. In these cases, a higher sweat rate can help to increase the concentration of these metabolites in the sweat, making them easier to detect and quantify. None of the five metabolites (glucose, sodium, CRP, uric acid, and cortisol) are consistently found in high concentrations in sweat[35]. Additionally, the relationship between sweat rate and the concentration of these metabolites in sweat is not well-established[32]. In general, however, it is essential to consider the impact of sweat rate on metabolite analysis carefully and to optimize experimental conditions accordingly.

Lower sweat rates may require longer collection times, which can affect the stability of some metabolites[147]. Additionally, sweat gland density, or the number of sweat glands per unit area of skin, can also affect metabolite analysis[142]. Different anatomical sites of the body have varying sweat gland densities, as shown in Fig. 2c, d, and some sites may produce sweat with higher concentrations of certain metabolites than others. Thus, it is recommended to optimize the anatomical site for sweat collection based on both sweat rate and sweat gland density to achieve more accurate metabolite analysis. Further research in this area is needed to improve the accuracy and reliability of sweat-based metabolite analysis.

While choosing an anatomical site that shows highest sweat rate in exercise or everyday setting may seem optimal for a wearable device, it is important to know that sweat rate significantly influences sodium levels[148,149], while there is no effect on glucose levels at physiologically normal sweat rates[61]. In addition to choosing an anatomical site that allows for non-invasive and continuous collection of a sufficient sweat sample, it is important to consider patient comfort and practicality when selecting the site.

**Hot versus cold**. The placement of wearables and sensing textiles for cold environments versus hot environments is of utmost importance in ensuring their efficacy and accuracy in collecting data. For instance, in cold environments, sweat production may decrease, and the sweat-based sensors may not work as effectively[150]. In these situations, it may be necessary to place the sensors in areas where the skin is warmer and more likely to produce sweat, such as the palms of the hands, as shown in Fig. 2. Placing the sensors in these areas can help ensure that there is enough sweat for the sensors to analyze.

In hot environments, excessive sweating can cause issues with sweat-based sensors[150]. In these conditions, it is essential to place the sensors in areas that are less prone to sweating, such as the soles of the feet as shown in Fig. 2, where they are less likely to be affected by excessive moisture. Heat can cause electronic sensors to malfunction or even stop working altogether for several reasons:

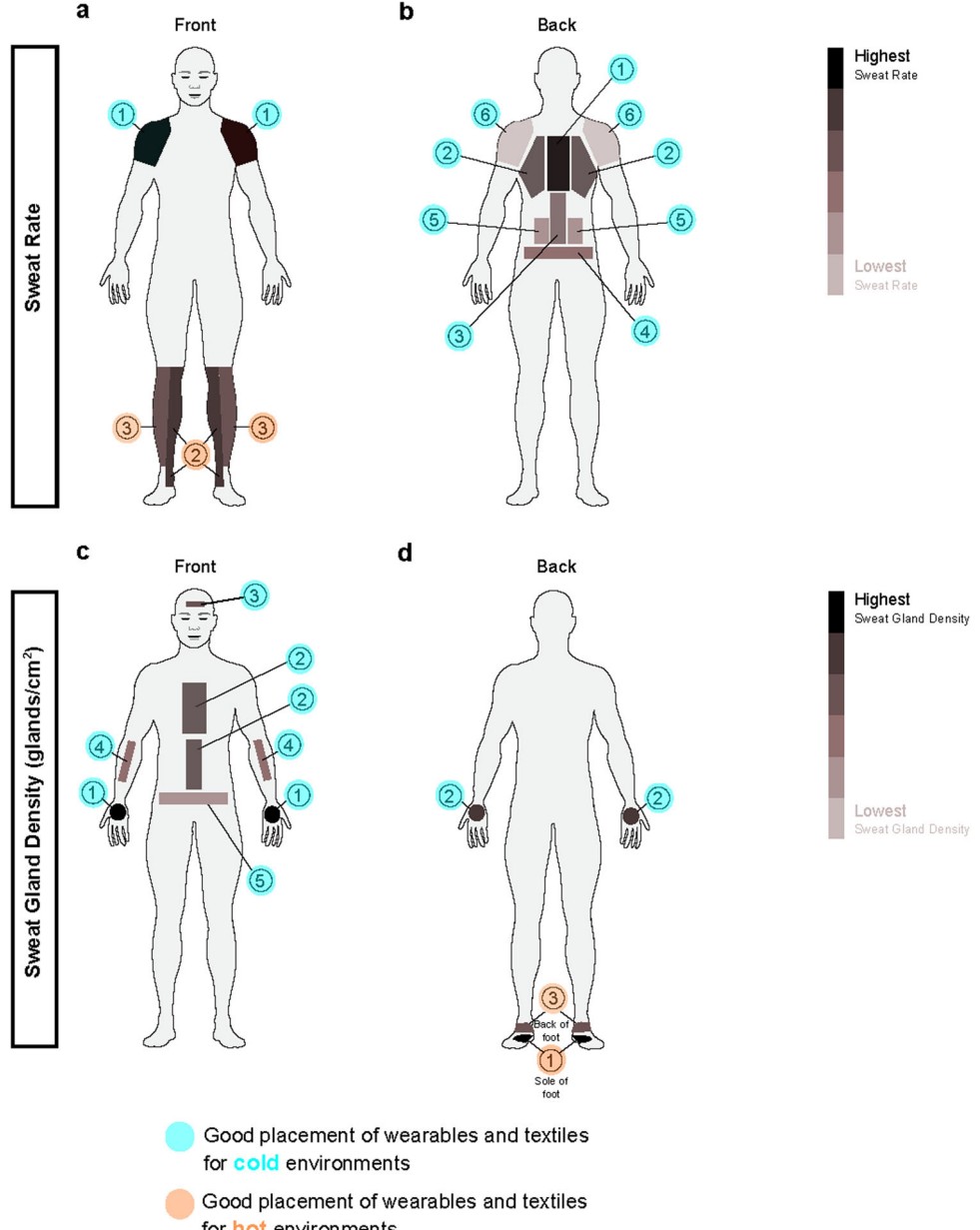

**Fig. 2 Ideal sensor placement on the skin for varying environmental conditions and sweat characteristics. a** Sweat rate ranked from highest to lowest on the front of the body, with 1 being the highest and 3 being the lowest. **b** Sweat rate ranked from highest to lowest on the back of the body, with 1 being the highest and 6 being the lowest. **c** Sweat gland density ranked from highest to lowest on the front of the body, with 1 being the highest and 5 being the lowest. **d** Sweat gland density ranked from highest to lowest on the back of the body, with 1 being the highest and 3 being the lowest. The sensor placement ranking was inspired by recently published sweat rate results[163] while the average sweat gland density illustration (glands/$cm^2$) on different areas of the body was based on these results[164,165]. The top body part is a good location for wearable placement in a cold environment, while the bottom legs are suitable for wearable placement in a hot environment[166].

- High temperatures can cause the electrical components inside the sensor to expand, leading to the deformation or breakage of internal structures[151,152].
- Heat can cause thermal noise, interfering with the sensor's signal[152,153].
- Excessive heat can cause the material used in the sensor's construction to degrade[152].

Therefore, placing the sensors in a well-ventilated area is essential to prevent heat buildup, which can cause them to malfunction.

Note that these sensors typically work by analyzing the sweat droplets that accumulate on the skin surface or are collected by the sensor. However, when the body overheats, sweat production can increase significantly, and this can cause issues for sweat-based sensors. Excessive sweating can create a barrier between the sensor and the metabolite being measured, leading to inaccurate readings. A high flow rate of the metabolite can help to increase the chances of the analyte molecules reaching the sensor surface and interacting with it, resulting in a more accurate measurement, especially for low concentration metabolite. Therefore, it is recommended that wearables and textiles be placed in areas where they are less likely to be exposed to external factors, temperature, humidity, and physical contact, which can affect their accuracy and reliability.

In both hot and cold environments, the design and materials used in the sweat-based sensors can also play a crucial role in their effectiveness. For example, in cold environments, it may be necessary to provide additional insulation around the sensors to prevent any temperature changes that could affect their accuracy. Sensors in contact with human skin are not likely to freeze in cold environments[154]. However, extreme cold can still affect the sensor's performance by decreasing sensitivity or causing other malfunctions[154]. In addition, if the sensor is located in an anatomical site exposed to cold air, such as fingers or toes, it may be more susceptible to cooling and require insulation to maintain its temperature and performance. Overall, while the risk of freezing may be low, it is still essential to consider the effects of cold temperatures on sensor performance and take appropriate measures to ensure accurate measurements. In hot environments, sensors may need to be made with materials that are resistant to moisture and heat, such as breathable fabrics that allow for better air flow and cooling[155].

**Passive versus active**. Several studies measured sweat passively[54,59,74,75,80,114], but the majority measured it actively, which involved exercise or physical activity. Two studies did not specify whether sweat collection was active or passive[50,57]. Most of the sweat samples analyzed were natural, although a few were artificial. Active sweating via physical activity may pose a challenge for less active Metabolic syndrome patients[125] who may find passive sweat collection more practical.

Furthermore, sweat collection methods (passive or active) may also determine the choice of wearable device placement. Active sweat collection can be used at any anatomical site, while passive sweat collection depends on a person's natural sweat rate (an uncontrollable variable)[32]. However, despite recent studies demonstrating that the composition of naturally produced and heat- or pharmaceutically-induced sweat is almost identical[156], sweat induction via iontophoretic or local heating methods may still be uncomfortable for many device wearers[157] who would prefer passive collection, if possible.

Although passive sweat collection may seem more user-friendly as it doesn't cause any discomfort, there are also some drawbacks to this method. Longer periods of sweat collection are required to obtain the same or smaller amounts of sweat compared to active induction methods[142]. This may not be an issue for stable aqueous analytes, but for metabolites that degrade quickly after excretion, prolonged collection time may lead to measurement inaccuracies. However, as sensors become increasingly miniaturized, smaller amounts of sweat are required for efficient analysis, potentially mitigating the problem of low sweat collection rates in passive mode[158].

At the same time, other methods of biofluid extraction, like reverse iontophoresis, can lead to generation of fluids that differ greatly in composition from sweat, e.g., interstitial fluid is extracted[159]. Heat- and exercise-induced sweat were also shown to differ in composition[160], which should be taken into account when designing possible context of the wearable device exploitation and how its reading are used in diagnosis.

In other words, passive sweat sensors rely on the natural flow of sweat from the skin, and environmental factors such as heat and humidity can affect the amount and flow of sweat produced, which can impact the accuracy of the sensor readings[161,162]. On the other hand, active sweat sensors, which use chemicals to stimulate the production of sweat, may not be as affected by weather conditions[161]. These types of sensors rely on a chemical reaction to trigger sweat production, and the resulting sweat is often more consistent in volume and flow rate, regardless of external environmental factors. However, it is important to note that the placement of any type of sensor should still be carefully considered to ensure optimal accuracy and reliable measurements. Factors such as the proximity to sweat glands, skin temperature, and skin type can all affect the performance of the sensor, and proper placement can help to minimize these potential sources of error.

**Conclusions and future directions**. Sweat-based wearable technologies have immense potential to gather crucial data on biomarkers related to metabolic syndrome and its associated diseases. These wearable devices can measure glucose, CRP, cortisol, and sodium levels, making them invaluable for patients at risk of developing diabetes and cardiovascular diseases. While the development of an integrated wearable textile capable of measuring all these biomarkers is feasible, more studies are necessary to validate the reliability and efficacy of sweat-based wearable devices. It is essential to establish the accuracy and effectiveness of these devices before they can be widely used in clinical settings to improve patient outcomes. Researchers developing new technologies able to monitor risk factors associated with metabolic syndrome must validate their findings by comparing them to gold standard measures. Validation should include comparisons to natural samples of the analyte in question. For instance, MARD is the current gold standard for measuring glucose in an individual's blood. For other biomarkers, a similar approach can be used in addition to artificial calibration curves (for instance, serial dilutions of $Na^+$ in laboratory environments).

Further research is urgently needed to better understand the composition of sweat in relation to active and passive modes, as well as other influential factors, in order to develop more effective and reliable wearable devices. Further research is also needed to determine the optimal and most convenient anatomical sites for assessing metabolites associated with metabolic syndrome, taking into account factors such as sweat rate and sweat gland density. Comparative studies should be conducted to evaluate the accuracy and efficacy of different sensor placements and inform the development of future wearable devices for metabolic syndrome monitoring.

Finally, there is a need for technological development focusing on identifying and improving the effectiveness of monitoring clinically identified risk factors associated with metabolic syndrome. Researchers focusing on fundamental technology development to validate monitoring technologies in a diversified population of participants should establish relationships with applied research groups that develop and complete larger clinical studies. These collaborations would inevitably improve such technologies by identifying any weaknesses while enhancing potential translation to and value for the general population.

**Reporting summary**. Further information on research design is available in the Nature Portfolio Reporting Summary linked to this article.

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

## Author contributions

L.L. drafted the initial versions of the manuscript and provided assistance with editing and revisions. M.E. led the study, designed the overall framework, offered critical feedback and input, provided guidance and team coordination, contributed to the creation of figures, and review development. A.S. contributed to the article screening process, assisted with editing, wrote parts of the discussion, provided critical feedback, and collaborated with ME on the creation of figures. T.C. contributed to the discussion section, offered advice on early version figures, and assisted with manuscript editing. C.A. contributed to a figure and a table in the early version of the paper, participated in discussions, and provided general editing support. C.M. formulated the fundamental concept of the study, provided senior supervisory guidance, edits, and feedback, and approved the final version of the manuscript. All authors reviewed and approved the final version of the manuscript for publication.

## Competing interests

The authors declare no competing interests.
