## [Peer Review File · Communications Engineering]

Reviewers' comments:

Reviewer #1 (Remarks to the Author):

It's a good review, it includes all the significant elements of a review paper. I suggest to accept it. Some minor suggestions: Add a architecture picture at the beginning of the paper to show the classification of the existing works. Add a perspective section at the end of the paper, to forecast the futher of the topic.

Reviewer #2 (Remarks to the Author):

The paper is well written and summarizes wearable technology for the metabolic syndrome assessment via sweat sensing. Various metabolic syndromes were summarized. The current status and prospects of using sweat sensors to detect biomarkers, including the choice of textile-based materials and major existing problems were discussed. Some minor issues need to be revised and improved.

1. Data on the concentration range of significant biomarkers of metabolic syndrome in sweat are missing. The differences in biomarker concentrations between normal and metabolic syndrome patients should be included.
2. It is suggested to include a few figures on wearable sensors for different biomarkers, so as to deliver the concept to broad audience in a more straightforward manner.
3. The choices of textile materials were not summarized. The article highlight textile-based sensor in some sections, while does not include discussion on the advantages of textiles, such as desirable skin contact, washability, and biocompatibility.
4. There is a lack of summary and discussion on sensor lifetime.
5. The citations in the article are sometimes confusing. For example, in Page 13, "There were a few studies that used minimally invasive microneedles to collect interstitial fluids from the epidermis. 19,61" . However, seems ref 61 is not relevant to the content. Please correct them.
6. There might be a lack of discussion of the effect of sweat rate or sweat volume on sensor testing. Active sweating, passive sweating, natural sweat production, and iontophoresis are mentioned. However, forward vs. reverse iontophoresis will produce different sweat rates and amounts of sweat, making a big difference in test results.

Reviewer #3 (Remarks to the Author):

Springer Nature – Communications Engineering
CommsEng-22-0068-T
Reviewer Comments – 31 May 2022

Metabolic syndrome assessment via wearable technologies: Where we are and how to move forward

Summary

This manuscript presents the results of a scoping review, exploring the detection of various metabolites relevant to the medical condition of Metabolic Syndrome using emerging sweat-based wearable

technologies. With the compiled information, the authors then discuss current limitations and future recommendations for designing future wearables to help diagnose, monitor and treat Metabolic Syndrome.

Overall Impression

The premise and content of this scoping review are interesting and relevant but the manuscript reads much too long. Significant editing of repetitive sections and restructuring of some of the sections, as indicated below in specific comments, would improve this manuscript.

There is inaccurate use of the word 'symptom' throughout the manuscript. Symptoms are defined in clinical medicine as what people/patients describe feeling or experiencing. Clinical symptoms generally cannot be seen and do not show up on medical testing. Clinical signs are objective findings that can be seen or measured. Metabolic Syndrome (MetSynd) is generally asymptomatic and that is precisely why it is such a great target medical condition for continuous wearable technology. The authors should consider replacing every mention of the word 'symptom' (of which there are many citations) with the word 'sign' or 'finding'. This may seem nit-picky, but the point of this review is to evaluate the role of these wearables in monitoring a medical condition, so using appropriate medical terminology is important.

Specific Comments

Title

The title needs clarification to be more specific to 'wearable sweat-based technologies'. All wearable technologies (e.g., interstitial fluid-ISF-microneedle based continuous glucose monitoring) currently available and relevant to MetSynd were not evaluated.

Abstract

Same comments apply, as detailed in overall impression – too repetitive (cardiovascular disease and type 2 diabetes need only be mentioned once) and inappropriate use of 'symptom'. Also, confusing information when describing % papers. The authors mention 20 studies that were included in the review. Yet in the next sentence they mention 'most of the papers (54%) were about sweat-based glucose sensors....' Did the authors mean 54% of all titles? All abstracts? All full text reviewed? This needs clarification. Maybe better to specify number of papers and not %.

Introduction

1st paragraph – reported prevalence of MetSynd approx. 30-40% of adults over age 65 – where? Worldwide? Europe? Canada? Why not reference all six different definitions of MetSynd here or later in introduction (contents of first 9 columns of Table 1), so that more emphasis can be made around the applicability of sweat-based wearables in the Results section? See below.

2nd paragraph – 2nd sentence is unclear: glucose [insulin intolerance (IRR) = 1.81]? What does this mean?

3rd paragraph – caution with language suggesting 'causation' vs 'correlation' – needs clarification – e.g., patients with MetSynd are more likely to exhibit sodium sensitivity, as opposed to currently written – patients with MetSynd have sodium sensitivity.

4th paragraph – 1st sentence – specify serum elevations of CRP; last sentence referring to hs-CRP (high

sensitivity CRP).

5th paragraph – Medically speaking, obesity is a medical condition, not a disease. Why are wristbands and watches considered part of smart clothing, but not rings (i.e. referring to popular Oura smart rings - <https://ouraring.com>)?

Aims

This section needs clarity and rewriting to better align with the findings of the study. Again, it wasn't all types of wearable technology applicable to MetSynd reviewed – only sweat-based ones. The entire paragraph could be condensed into 1-2 sentences.

Methods

The current order of paragraphs seems disorganized to me. Why not reflect the order that the study search was done: existing paragraph 1 (search databases, search dates – condensed into one sentence without repetition), followed by paragraph 4 (search terms), then paragraph 2 (inclusion criteria), then paragraph 3 (exclusion criteria); and then data screening and data extraction combined into one small paragraph, not separate sections.

When did the authors include/exclude human vs animal studies? Surely, this should be mentioned.

Results

The section 'Targets for sweat-based metabolic syndrome assessment' could be streamlined better with the contents of Table 1 (Different clinical definitions of metabolic syndrome) if Table 1 were introduced earlier in the paper (e.g. in the Introduction). It seems outside the scope of this review to evaluate the clinical definitions of MetSynd. Instead, if I understand correctly, it is to evaluate how sweat-based metabolites might contribute to better detecting and monitoring (not defining) the clinical syndrome of MetSynd – essentially the results are only detailed in the final column of Table 1? Why not focus on the findings accordingly?

The rest of the Results section (a paragraph for each identified biomarker target) and Table 2 flow well.

Discussion

This section seems exceedingly long.

The first paragraph needs a summary of the main findings, and not a repetition of the aims/ objectives.

'Validity and reliability' section - define all acronyms and avoid repetitive sentences and phrases within paragraphs.

'An integrated wearable sensor for monitoring metabolic syndrome section' – again very wordy with repetitive information. 2nd paragraph – what is meant by the sentence, "One study found that, overall, risk calculation was accurate and its system was able to determine which patients needed medical treatment rather than medication"? Isn't medication part of medical treatment? Some further explanation is missing here. 6th paragraph – again, why no mention of the wearable oura ring when discussing the current market? 7th paragraph – redundant information, consider omitting.

'Optimal sensing material for monitoring metabolic syndrome' section reads more concisely but it is outside my area of expertise.

'Recommendations' section needs focus. Advise omission of #1 – redundant. Recommendations #2 and #3 could be broken down into several briefer recommendations, each with one main focus. This is a summary section, not a time to introduce examples, new material, etc.

The conclusion section is clear and concise.

Reviewer #1 (Remarks to the Author):

It's a good review, it includes all the significant elements of a review paper. I suggest to accept it. Some minor suggestions: Add a architecture picture at the beginning of the paper to show the classification of the existing works. Add a perspective section at the end of the paper, to forecast the futher of the topic.

Author response: We thank the reviewer for the positive feedback and for accepting the work for publication.

Author action 1: A perspective section is added at the end of the paper and reads as follows.

In Perspective: A Future integrated wearable sensor for monitoring metabolic syndrome

The future of wearable sensors should involve the development of an integrated wearable piece of smart clothing that can collect data on multiple relevant clinical parameters and risk factors. Ideally, smart clothing with integrated sensors should be made of comfortable fabric that can be easily worn and allow for data from the MetS patient to be passively collected.

A future wearable for monitoring metabolic syndrome could also collect data on cortisol levels to mitigate the risk in patients. Researchers have found that patients with metabolic syndrome have higher amounts of an enzyme (11B-HSD1) that converts the inactive stress hormone cortisone to cortisol in the liver,^{82,83} and high cortisol levels are associated with an increased risk of premature mortality from cardiovascular disease (CVD). It has been previously theorized that signs associated with metabolic syndrome—including adiposity, high blood pressure, and insulin resistance—could be reduced if cortisol levels are controlled.⁸⁴ Research indicates that patients with high cortisol levels (measured via urinary excretion) were five times more likely to die from CVD over a six-year period⁸⁵ than their counterparts with lower cortisol levels.

Furthermore, biosensors that collect data on fibrinogen and cholesterol are needed to complete a comprehensive mobile wearable that will provide the best clinical diagnostics. A future wearable should also measure abdominal obesity via waist circumference as these indicators are highly relevant to the syndrome, highlighted in Table 1.

In addition, a future sensor for MetS could also measure physical activity. Specifically, low levels of physical activity may result in obesity and being overweight,⁸⁶ high blood pressure,⁸⁷ and insulin intolerance,⁸⁸ all of which are strongly associated with metabolic syndrome. The risk of MetS is reduced with increasing levels of physical activity and reduced sedentary behaviour^{89,90}.

Author action 2: The architecture of the introduction is improved.

Reviewer #2 (Remarks to the Author):

The paper is well written and summarizes wearable technology for the metabolic syndrome assessment via sweat sensing. Various metabolic syndromes were summarized. The current status and prospects of using sweat sensors to detect biomarkers, including the choice of textile-based materials and major existing problems were discussed. Some minor issues need to be revised and improved.

Author response: We thank the reviewer for the overall positive feedback for recognizing the merit of our work.

Author action: None.

1. Data on the concentration range of significant biomarkers of metabolic syndrome in sweat are missing. The differences in biomarker concentrations between normal and metabolic syndrome patients should be included.

Author response: Thank you for your valuable suggestion! Indeed, the blood levels of these biomarkers are considered the gold standard in diagnosing metabolic diseases. Thus, drawing a comparison between those blood and sweat levels is essential. And while many studies show that several biomarkers show strong correlations in sweat vs. blood concentrations, some show the inverse. Notable, sodium levels in sweat barely correspond to serum levels. However, sweat sodium concentration, independently from serum, is attracting more attention from the biomedical community as an independent biomarker. While we now provide literature references that show correlations between sweat and blood biomarker levels, there are no consolidated reports on these specifically for patients diagnosed with metabolic syndrome. While it can be assumed that these levels correspond to respective blood threshold values, concluding so in a review paper is entirely speculative. We are sure that, in the near future, sweat science will be able to elucidate these concentration values for sweat biomarkers as well.

Author action: The part of the manuscript discussing the measurement of essential biomarkers now includes text on correlation with blood levels for each:

“It should be noted that glucose levels in sweat correlate strongly to blood glucose concentration [doi.org/10.1089/dia.2011.0262], which is considered a diagnostic value for diabetes and metabolic syndrome.”

“While recent studies show that sweat sodium concentration is almost certainly independent of its blood concentration [10.1007/s00421-020-04562-8], this value for sweat is now considered a good diagnostic value by itself, along with sweat chloride levels [10.1258/000456307779596011].”

“Several studies on relationships between levels of inflammatory proteins in sweat and blood have shown strong correlations [10.1016/j.jim.2006.07.011 + j.biopsych.2008.05.035], further supporting the feasibility of measurement of these biomarkers in sweat for diagnostic applications.”

“While uric acid in sweat was found to be non-correlated to serum levels [PMID: 12817713.], it starts to be considered an important biomarker itself [10.1038/s41587-019-0321-x].”

“It is important to underline that sweat cortisol levels were previously found to correlate strongly to serum levels [10.1016/j.matt.2020.01.021], making non-invasive measurements by wearable devices a promising tool for diagnosing metabolic syndrome.”

2. It is suggested to include a few figures on wearable sensors for different biomarkers, so as to deliver the concept to broad audience in a more straightforward manner.

Author response: We see the value of this excellent suggestion. We feel that Figure 3 is sufficient to show how different biomarkers can be measured in one integrative sensing patch. Adding multiple figures on wearable sensors for other biomarkers may distract the readers from the paper's main message; therefore, we respectfully did not add more figures.

Author action: None.

3. The choices of textile materials were not summarized. The article highlight textile-based sensor in some sections, while does not include discussion on the advantages of textiles, such as desirable skin contact, washability, and biocompatibility.

Author response: We have added a discussion to the introduction highlighting why and how textile-based devices are advantageous as sensing devices.

Author action: We added the following sentences to address this point.

“Textile-based devices (i.e., clothing) have the potential to decrease the barrier of access to biosignal monitoring since they are familiar to essentially all populations, sit on/close to the body, and can be employed on all parts of the body, which may increase acceptance, enable capture of different signal types, and expand accessible areas of the body for monitoring, respectively. In addition, textile-based sensors should ideally possess similar characteristics and usability to traditional textiles/clothing to allow ease of use and direct integration into daily lives, such as ensuring biocompatibility to allow long-term use (and reduce the chance of irritation often caused by medical devices) and compatibility with common cleaning/washing machines to ensure devices can be (re)used daily.”

4. There is a lack of summary and discussion on sensor lifetime.

Author response: Thank you for your valuable suggestion! We have analyzed the works selected for the scoping review regarding reported sensors' stability and lifetime. This information is now provided in Table 2.

Author action: Added a new column containing data on sensor lifetimes and stability to Table 2.

5. The citations in the article are sometimes confusing. For example, in Page 13, "There were a few studies that used minimally invasive microneedles to collect interstitial fluids from the epidermis. 19,61" . However, seems ref 61 is not relevant to the content. Please correct them.

Author response: We thank the reviewer for pointing this out.

Author action: References are fixed.

6. There might be a lack of discussion of the effect of sweat rate or sweat volume on sensor testing. Active sweating, passive sweating, natural sweat production, and iontophoresis are mentioned. However, forward vs. reverse iontophoresis will produce different sweat rates and amounts of sweat, making a big difference in test results.

Author response: Agreed, the manuscript indeed lacked discussion on these topics. The sweat rate is undoubtedly one of the most critical parameters in sweat analysis. Moreover, the sweat rate for different activities and methods of inducing secretion will differ significantly. This is an aspect that is not always considered in developing various chemosensors. While it is a fascinating and vital topic, it is as extensive. At the same time, our manuscript is mainly devoted to perspective devices for diagnosing metabolic syndrome. Nonetheless, we have somewhat expanded the part of the article on optimal anatomical sites and collection methods to provide some pointers toward sweat rate and composition differences that may be important for future developments.

Author Action: Expanded the discussion on optimal anatomical site and sweat collection methods for monitoring metabolic syndrome.

Reviewer #3 (Remarks to the Author):

Springer Nature – Communications Engineering

CommsEng-22-0068-T

Reviewer Comments – 31 May 2022

Metabolic syndrome assessment via wearable technologies: Where we are and how to move forward

Summary

This manuscript presents the results of a scoping review, exploring the detection of various metabolites relevant to the medical condition of Metabolic Syndrome using emerging sweat-based wearable technologies. With the compiled information, the authors then discuss current limitations and future recommendations for designing future wearables to help diagnose, monitor and treat Metabolic Syndrome.

Author response: We thank the reviewer for the positive feedback.

Author action: None.

Overall Impression

The premise and content of this scoping review are interesting and relevant but the manuscript reads much too long. Significant editing of repetitive sections and restructuring of some of the sections, as indicated below in specific comments, would improve this manuscript.

Author response: We thank the reviewer for the positive feedback.

Author action: None.

There is inaccurate use of the word 'symptom' throughout the manuscript. Symptoms are defined in clinical medicine as what people/patients describe feeling or experiencing. Clinical symptoms generally cannot be seen and do not show up on medical testing. Clinical signs are objective findings that can be seen or measured. Metabolic Syndrome (MetSynd) is generally asymptomatic and that is precisely why it is such a great target medical condition for continuous wearable technology. The authors should consider replacing every mention of the word 'symptom' (of which there are many citations) with the word 'sign' or 'finding'. This may seem nit-picky, but the point of this review is to evaluate the role of these wearables in monitoring a medical condition, so using appropriate medical terminology is important.

Author response: We thank the reviewer for the valuable suggestion.

Author action: "symptoms" has been replaced with "clinical signs" throughout.

Specific Comments

Title

The title needs clarification to be more specific to 'wearable sweat-based technologies'. All wearable technologies (e.g., interstitial fluid-ISF-microneedle based continuous glucose monitoring) currently available and relevant to MetSynd were not evaluated.

Author response: Agreed, we thank the reviewer for the valuable suggestion.

Author action: The title has been modified to reflect "sweat-based" technology.

Abstract

Same comments apply, as detailed in overall impression – too repetitive (cardiovascular disease and type 2 diabetes need only be mentioned once) and inappropriate use of 'symptom'. Also, confusing

information when describing % papers. The authors mention 20 studies that were included in the review. Yet in the next sentence they mention ‘most of the papers (54%) were about sweat-based glucose sensors...’ Did the authors mean 54% of all titles? All abstracts? All full text reviewed? This needs clarification. Maybe better to specify number of papers and not %.

Author response: We thank the reviewer for the positive feedback.

Author action: Repetition has been removed from the abstract and the word “symptom” has not been used. The number of articles and the percentage meaning has been clarified.

Introduction

1st paragraph – reported prevalence of MetSynd approx. 30-40% of adults over age 65 – where? Worldwide? Europe? Canada? Why not reference all six different definitions of MetSynd here or later in introduction (contents of first 9 columns of Table 1), so that more emphasis can be made around the applicability of sweat-based wearables in the Results section? See below.

Author response: We thank the reviewer for asking important questions.

Author action: More information has been provided on the prevalence of metabolic syndrome as follows.

“The reported prevalence of metabolic syndrome is high, approximately 30–40% of adults over the age of 65 suffer from it.⁶ A recent systematic review found that 23.7% of adults worldwide living with type I diabetes mellitus have metabolic syndrome⁷. Similarly high estimates were found among obese adults in Europe, with 24–65% of obese women having metabolic syndrome and 43–78% of obese men⁸. There has been a steady rise in the global prevalence of the syndrome over the last decades,⁹ and the trends have shown a higher prevalence in older age groups (e.g., over age 70 years)¹⁰.”

2nd paragraph – 2nd sentence is unclear: glucose [insulin intolerance (IRR) = 1.81]? What does this mean?

Author response: We thank the reviewer for the point this out.

Author action: IRR has been explained in the second paragraph of the introduction. It is the Incidence Rate Ratio.

3rd paragraph – caution with language suggesting ‘causation’ vs ‘correlation’ – needs clarification – e.g., patients with MetSynd are more likely to exhibit sodium sensitivity, as opposed to currently written – patients with MetSynd have sodium sensitivity.

Author response: We thank the reviewer for pointing this out.

Author action: The language has been refined to ensure that it does not imply causation.

4th paragraph – 1st sentence – specify serum elevations of CRP; last sentence referring to hs-CRP (high sensitivity CRP).

Author response: We thank the reviewer for the valuable suggestion. CRP concentrations have been listed, including what are the general levels in people with MetS and a reference on concentration in high sensitive CRP individuals has been added. The first sentence refers to a review on the topic which listed different concentrations, hence we refer to the literature.

Author action: We added the following sentences to address this point.

“A proinflammatory state, recognized clinically by elevations of C-reactive protein (CRP), is commonly present in persons with metabolic syndrome.^{16,17} Research in men has found that concentrations above 3 mg/dl significantly increase the risk of MetS by 3 fold relative to men with less than 1 mg/ dl of CRP¹⁷. Specifically, CRP has been linked to the development of hypertension in individuals who previously had normal blood pressure, and concentrations above 3 mg/ dl increase the risk of CVD¹⁸⁻²¹. High CRP levels have also been found to be associated with an increased risk of developing insulin resistance and type 2 diabetes.^{17,22} Additionally, patients with existing metabolic syndrome are at a much higher risk of developing cardiovascular disease when high sensitive C-reactive protein levels are elevated >3.0 mg/l.¹⁷”

5th paragraph – Medically speaking, obesity is a medical condition, not a disease. Why are wristbands and watches considered part of smart clothing, but not rings (i.e. referring to popular Oura smart rings - <https://ouraring.com>)?

Author response: Agreed. We thank the reviewer for pointing this out.

Author action: We added this following sentences to address this point.

“Thus, there is a need to develop effective and scalable interventions for metabolic syndrome.^{1,2} In recent years, mobile health (mHealth) technology has emerged as a medium for promoting behavior changes and reducing lifestyle-related risk factors associated with chronic conditions, such as obesity.²³ mHealth technology also has the potential to assess metabolic syndrome and its associated risk factors and clinically relevant parameters via the use of emerging wearable smart clothing—textiles, wristbands, rings²⁴(e.g., Oura ring)²⁵, and smartwatches—that collect biomedical data from subjects with different health conditions.²⁶⁻²⁸”

Aims

This section needs clarity and rewriting to better align with the findings of the study. Again, it wasn't all types of wearable technology applicable to MetSynd reviewed – only sweat-based ones. The entire paragraph could be condensed into 1 -2 sentences.

Author response: We thank the reviewer for the excellent tip.

Author action: We modified the aims section and now it reads as follows.

“This study aimed to review what types of sweat-based wearable technology have been developed to measure one or more clinical signs of metabolic syndrome. We also aimed to make recommendations for future technological development in this field.”

Methods

The current order of paragraphs seems disorganized to me. Why not reflect the order that the study search was done: existing paragraph 1 (search databases, search dates – condensed into one sentence without repetition), followed by paragraph 4 (search terms), then paragraph 2 (inclusion criteria), then paragraph 3 (exclusion criteria); and then data screening and data extraction combined into one small paragraph, not separate sections.

When did the authors include/exclude human vs animal studies? Surely, this should be mentioned.

Author response: We thank the reviewer for the valuable suggestions. The methods section paragraphs 1-4 have been re-organized to reflect the order suggested by the reviewer. The last two paragraphs have been condensed into one as well. We also added a note about the exclusion of animal studies.

Author action: We modified the methods section, and now it reads as follows.

“A scoping review of PubMed was undertaken in September 2021 to search for relevant papers to answer our research questions. Google Scholar was also searched for additional studies. The search was carried out over the past decade (1st of September 2011 to 1st of September 2021). Our search terms included a combination of medical subject terms and free text keywords, such as “metabolic syndrome OR diabetes OR abdominal obesity OR hypertension OR glucose OR inflammation AND wearable devices OR mHealth” (see Supplementary 1 for the PubMed search strategy).

We broadly included all interventional, pilot, and proof-of-concept wearable technology studies with human participants with metabolic syndrome or with participants with one or more clinical parameters or signs related to metabolic syndrome. These included studies concerning individuals with diabetes, studies monitoring glucose or sodium levels in the body, and studies focusing on hypertension, inflammatory markers, or physical activity indicators. The studies of the greatest relevance concerned emerging technologies that focused on fabric-type wearables—textile-based fabrics, smart clothing,

wristbands, and other fabric-based wearable sensors—that collected sweat or other bodily fluids, including interstitial fluid (ISF), from the epidermis in a non-invasive way.

We excluded studies not published in English and studies older than 10 years. Animal studies were also excluded. Review papers, conference papers, and papers without full-text availability were also excluded. During screening abstracts and the full texts of studies, our exclusion criteria were based on biomarkers investigated and the measurement method. Concerning the first criterion, papers that did not explore at least one of our target biomarkers in sweat were excluded. For example, we did not include studies that measured sweat rates. The second inclusion criterion—the measurement method—dictated how the target biomarker was quantified. Based on this criterion, studies that indirectly used machine learning to estimate a biomarker used non-sweat-based sensors or implemented manual recordings of biomarkers were excluded. Also not included were studies that focused on smartphone sensors without using a fabric-type wearable (i.e., a wristband, smart clothing, or other textile-based fabric). Studies involving well-established technologies that were not novel and emerging, such as personal digital assistants (PDAs), were also excluded. Finally, cortisol measurement was included only if another biomarker directly relevant to metabolic syndrome (e.g., glucose) was also included in the study.

Data screening was undertaken sequentially by two reviewers (LNL and ME). We first screened titles, followed by abstracts, for relevance, screening against our inclusion and exclusion criteria. We then retrieved the full texts of the articles that met our inclusion criteria. If the two reviewers could not reach a consensus about an article, a third reviewer (AS) broke the tie. Descriptive data on studies, technology, clinical relevance, results, and validity were extracted and summarized in tabular format.”

Results

The section ‘Targets for sweat-based metabolic syndrome assessment’ could be streamlined better with the contents of Table 1 (Different clinical definitions of metabolic syndrome) if Table 1 were introduced earlier in the paper (e.g. in the Introduction). It seems outside the scope of this review to evaluate the clinical definitions of MetSynd. Instead, if I understand correctly, it is to evaluate how sweat-based metabolites might contribute to better detecting and monitoring (not defining) the clinical syndrome of MetSynd – essentially the results are only detailed in the final column of Table 1? Why not focus on the findings accordingly?

Author response: Agreed.

Author action: The results section has been modified as requested by the reviewer. Table 1 has been moved up into the introduction. As the reviewer felt that the section on the applicability of the sensors to the definition was out of the scope of the results, this information was moved to the introduction as a general overview.

The rest of the Results section (a paragraph for each identified biomarker target) and Table 2 flow well.

Author response: We thank the reviewer for the positive feedback on the rest of the Results section.

Author action: None.

Discussion

This section seems exceedingly long.

The first paragraph needs a summary of the main findings, and not a repetition of the aims/ objectives. 'Validity and reliability' section - define all acronyms and avoid repetitive sentences and phrases within paragraphs. 'An integrated wearable sensor for monitoring metabolic syndrome section' – again very wordy with repetitive information.

Author response: We thank the reviewer for the valuable suggestions.

Author action 1: The discussion was shortened and modified per the reviewer's request. All acronyms have now been defined. There is one sentence that summarizes the aim before diving into the details of the discussion, but it is not detailed and flows well with the organization of the section.

Author action 2: The redundant information was removed from the "an integrated sweat-based sensor" paragraph. The paragraph that the reviewer suggested removing was removed. The section on cortisol and physical activity was shortened.

2nd paragraph – what is meant by the sentence, "One study found that, overall, risk calculation was accurate and its system was able to determine which patients needed medical treatment rather than medication"? Isn't medication part of medical treatment? Some further explanation is missing here.

Author response: We thank the reviewer for the valuable suggestion.

Author action: We removed this sentence for clarity.

6th paragraph – again, why no mention of the wearable oura ring when discussing the current market?

7th paragraph – redundant information, consider omitting.

Author response: We thank the reviewer for the great tip.

Author action: We did not mention the Oura ring in the discussion as we mentioned it in the introduction as per the reviewer's earlier advice. We removed the broad information on other technology as we agreed that some things were beyond the scope of the discussion. We worked on ensuring that the discussion was simplified and shortened.

'Recommendations' section needs focus. Advise omission of #1 – redundant. Recommendations #2 and #3 could be broken down into several briefer recommendations, each with one main focus. This is a summary section, not a time to introduce examples, new material, etc.

Author response: We thank the reviewer for the great tip.

Author action: The recommendations section was edited per the reviewer's suggestions. We removed the first recommendation as requested by the reviewer. We further broke the section down into several more minor recommendations as suggested by the reviewer, and we did not add more detailed examples of new materials as requested.

The conclusion section is clear and concise.

Author response: We thank the reviewer for the positive feedback on the conclusion section.

Author action: None.

Reviewers' comments:

Reviewer #1 (Remarks to the Author):

It's acceptable.

Reviewer #2 (Remarks to the Author):

The authors have revised the manuscript according to the comments and it can be considered for publication. A minor suggestions as follow: In the discussion of sweat biomarkers correlation with blood, please be very careful on indication 'strong correlations'. While some reports indicate some conclusions, they have not yet been validated in comprehensive clinical study or applications.

Reviewer #3 (Remarks to the Author):

This revised manuscript is now much improved in its content, structure and readability. All of my concerns have been adequately addressed.

-Reviewer #3

Reviewer #4 (Remarks to the Author):

The authors discuss the concepts of metabolic syndrome (MetS) while describing the importance of prevention and monitoring of such. They discuss several sweat sensors in the literature that were already developed for some analytes correlated with MetS and how it would be possible to combine them to develop a wearable sweat MetS sensor. This reviewer recognizes the importance of this discussion and recommends the publication of the work so it can bring awareness to such a topic. However, the current version, despite being well-written and organized, is somewhat superficial, and predictable, and it does not bring much new insight from the experts. Even though some information is not yet available, the authors could give their opinion regarding the same. For example, lipidic levels in sweat have not been demonstrated, however, their levels in the blood are in the range of mM so, based on similar molecules, sweat levels should be 1000x less, what makes their detection very difficult, on the other hand, a good correlation with blood is expected for these molecules once they are lipophilic. Moreover, locations with a higher density of sebaceous glands would favor the detection. This is only one example of the level of discussion this reviewer was expecting for the whole manuscript. It is simple to write that the information is not available and not discuss it any further, the work would be more helpful for the readers if the authors could input more of their expert knowledge. The work overall is great and very important.

Please find below some minor suggestions:

1. "While simple skin cleaning before a measurement can be feasible in a clinical setting, the same cannot be done throughout the day while using a mobile sensor applied to the skin" Please comment on why the skin must be constantly cleaned while using the wearable device. Please give an example of such a case.

2. "Sensing modalities that can be miniaturized and used in wearable devices also exhibit quite limited shelf-life, according to several studies which remains a significant focus of research in wearable glucose sensors." Please make clear what are these sensing modalities

3. "We identified one study that measured uric acid levels—in addition to glucose and sodium using a multi-sensor wearable device." Why not include the following work: Yang, Yiran, et al. "A laser-engraved wearable sensor for sensitive detection of uric acid and tyrosine in sweat." *Nature biotechnology* 38.2 (2020): 217-224.

4. "We identified proof-of-concept studies that have been undertaken to monitor glucose, sodium, CRP, and blood pressure levels in this review." Was blood pressure reviewed? Please revise

5. "Thus, a future integrated sensor for metabolic syndrome would ideally collect multiple data on levels of glucose, sodium, hypertension, CRP, uric acid, cortisol, and physical activity." Please explain why cholesterol and blood pressure were not included

6. Please discuss what should be considered to choose a location. does the location matter for the measured data? What are the criteria for choosing the best location? (Please include a discussion where the authors think it fits best)

7. "Aging of the materials needs to be considered, whereby signals are no longer transmitted efficiently" please explain further. what is the meaning of aging of the material?

8. "For example, rapid fouling in sweat remains a considerable problem in fine techniques like aptasensing" What are fine techniques? Why aptasensors are part of it? Please explain

9. "crosstalk may become an issue that can be solved through appropriate data acquisition electronics" What about chemical cross-talking? For example, in your proposed sensor in figure 2, there might be chemical crosstalking between glucose and uric acid if the measurement is based on hydrogen peroxide coming from both reactions. Please discuss.

10. Figure 3, please discuss further the “cold and hot environment” placements.

11. “However, despite recent studies demonstrating that the composition of naturally produced and heat- or pharmaceutically-induced sweat is almost identical, sweat induction via iontophoretic or local heating methods may still be uncomfortable for many device wearers who would prefer passive collection, if possible” Please give examples and discuss passive sweat collection

12. Please consider including sensing technologies that are still not wearable as an opportunity to create the most comprehensive MetS wearable. The current perspective section is just a repetition of the previous discussion. In the future, new technologies should be proposed. For example, there are many enzymatic sensors for measuring cholesterol and triglycerides, there are epidermal sensors able to measure blood pressure, there are commercial devices able to measure the fat under the skin using skin resistance, etc. Figure 2 should be completely modified to include a comprehensive device and a better representation of the sensor. This perspective should guide and inspire future researchers, thus, the future device for MetS should be complete at least in concept. Try including how the signal would be transmitted, and utilized by the wearer or caregiver, including a diagram, etc. The figure should include, all the parameters considered important to be monitored by a MetS wearable device.

Reviewer #5 (Remarks to the Author):

This review paper shows that metabolic syndrome can be evaluated with wearable sweat sensors. I think this is an important paper that shows the applicability of many previously developed wearable sweat sensors. The author organized metabolic diseases well and classified them according to the substance to analyze wearable sensors. I think this paper is systematically written and will be of great help to readers. If possible, organizing the representative images of wearable sweat sensors, their adhesion to the skin, and how to collect sweat through pictures will help readers understand easily.

Reviewer #2

The authors have revised the manuscript according to the comments and it can be considered for publication. A minor suggestions as follow: In the discussion of sweat biomarkers correlation with blood, please be very careful on indication 'strong correlations'. While some reports indicate some conclusions, they have not yet been validated in comprehensive clinical study or applications.

Author response: Thank you for your valuable feedback and accepting the paper for publication.

Author Action: We remove the word “strong” to address the raised point.

Reviewer #4

1. “While simple skin cleaning before a measurement can be feasible in a clinical setting, the same cannot be done throughout the day while using a mobile sensor applied to the skin” Please comment on why the skin must be constantly cleaned while using the wearable device. Please give an example of such a case.

Author response: The phrase in the manuscript was not clearly written. We have meant that skin contamination leads to inaccurate readings from sweat, which can be amended using skin washing before sample collection (10.1152/jappphysiol.01437.2010). In case of skin-adhered wearable devices, however, it is obviously impossible to wash the skin underneath. This is a problem since contamination and biofouling lead to reduction in measurement quality. We have now expanded the paragraph in question with brief discussion of this matter and perspectives on how the issue of sensor contamination can be leveled off with new materials.

Author Action: We clarified this point by adding the following:

A recent study⁶⁹ looked into the composition and mechanisms of contaminants in sweat, which can accumulate on the surface of sweat-based sensors and interfere with their accuracy in measuring mineral levels. These contaminants can include proteins and lipids, which can result in inaccurate measurements and reduce the reliability of the sensor. This buildup of contaminants on the sensor's surface is referred to as "biofouling," and it can occur over time due to various factors, such as the composition of the sweat, the design of the sensor, and the duration of use. Although cleaning the skin before measuring can help in a clinical setting, it is not practical to do so regularly during mobile sensor use. To address this issue, researchers are exploring alternative methods such as selective membranes, anti-fouling layers, and enzyme stabilizers, which can prevent contaminants from sticking to the sensor surface and improve the sensor's accuracy and reliability⁷⁰.

2. “Sensing modalities that can be miniaturized and used in wearable devices also exhibit quite limited shelf-life, according to several studies which remains a significant focus of research in wearable glucose sensors.” Please make clear what are these sensing modalities

Author response: Thank you for your feedback. The phrase in question pertained specifically to enzymatic glucose sensors, which typically have a shorter shelf life compared to inorganic non-enzymatic sensors. We have now expanded and revised the paragraph to provide a clearer explanation of the various sensing modalities.

Author Action: We clarified this point by adding the following:

The sensing modalities commonly used in wearable devices that can be miniaturized rely on enzymatic reactions, which offer high selectivity and efficient analytical performance. However, enzyme-based sensors are known to have limited shelf-life, as demonstrated by several studies^{71,72}, which highlights the need for further research in this area, particularly for wearable glucose sensors. To address this issue, current research is focused on exploring non-enzymatic pathways for glucose detection, which offer the promise of extended shelf-lives⁷³.

3. "We identified one study that measured uric acid levels—in addition to glucose and sodium using a multi-sensor wearable device." Why not include the following work: Yang, Yiran, et al. "A laser-engraved wearable sensor for sensitive detection of uric acid and tyrosine in sweat." *Nature biotechnology* 38.2 (2020): 217-224.

Author response: Thank you for your suggestion. This paper was not included as the creating a technology to measure both uric and tyrosine is out of scope. However, the paper was referenced in our paper.

Author Action: None.

4. "We identified proof-of-concept studies that have been undertaken to monitor glucose, sodium, CRP, and blood pressure levels in this review." Was blood pressure reviewed? Please revise

Author response: Thank you for pointing this out.

Author Action: The word blood pressure was removed.

5. "Thus, a future integrated sensor for metabolic syndrome would ideally collect multiple data on levels of glucose, sodium, hypertension, CRP, uric acid, cortisol, and physical activity." Please explain why cholesterol and blood pressure were not included

Author response: Thank you for pointing this out. We are looking for five specific biomarkers to assess the metabolic syndrome.

Author Action: Cholesterol and hypertension were removed from the sentence and it reads as follows.

"Thus, a future integrated sensor for metabolic syndrome would ideally collect multiple data on levels of glucose, sodium, CRP, uric acid, and cortisol."

6. Please discuss what should be considered to choose a location. does the location matter for the measured data? What are the criteria for choosing the best location? (Please include a discussion where the authors think it fits best)

Author response: Thank you for your valuable suggestion.

Author Action 1: We clarified this point by adding a new figure, Figure 2.

Author Action 2: We clarified this point by adding the following.

Studies^{149,150} have shown that sweat rates vary significantly across the body, as evidenced by thermal mapping and sweat collection from specific areas of the body, as shown in Figures 3a and 3b. This difference in sweat rates can affect the accuracy of metabolite analysis, with higher sweat rates leading to more accurate results for some low-concentration metabolites. However, lower sweat rates may require longer collection times, which can affect the stability of some metabolites. Additionally, sweat gland density, or the number of sweat glands per unit area of skin, can also affect metabolite analysis. Different anatomical sites of the body have varying sweat gland densities, as shown in Figures 2c and 2d, and some sites may produce sweat with higher concentrations of certain metabolites than others. Thus, it is recommended to optimize the anatomical site for sweat collection based on both sweat rate and sweat gland density to achieve more accurate metabolite analysis. Further research in this area is needed to improve the accuracy and reliability of sweat-based metabolite analysis.

Author Action 3: We clarified this point by adding the following.

The placement of wearables and sensing textiles for cold environments versus hot environments is of utmost importance in ensuring their efficacy and accuracy in collecting data. For instance, in cold environments, sweat production may decrease, and the sweat-based sensors may not work as effectively. In these situations, it may be necessary to place the sensors in areas where the skin is warmer and more likely to produce sweat, such as the palms of the hands, as shown in Figure 2. Placing the sensors in these areas can help ensure that there is enough sweat for the sensors to analyze

In hot environments, excessive sweating can cause issues with sweat-based sensors. In these conditions, it is essential to place the sensors in areas that are less prone to sweating, such as the soles of the feet as shown in Figure 2, where they are less likely to be affected by excessive moisture. Additionally, the sensors should be placed in a well-ventilated area to prevent heat buildup, which can cause them to malfunction.

Note that these sensors typically work by analyzing the sweat droplets that accumulate on the skin surface or are collected by the sensor. However, when the body overheats, sweat production can increase significantly, and this can cause issues for sweat-based sensors. Excessive sweating can lead to a buildup of sweat on the sensor, which can interfere with the accuracy of the readings. Additionally, high temperatures can cause the sensor to malfunction or even stop working altogether. Therefore, it is recommended that wearables and textiles be placed in areas where they are less likely to be exposed to external factors, temperature, humidity, and physical contact, which can affect their accuracy and reliability such as the shoulder area or the back of the hand, as shown in Figure 2.

In both hot and cold environments, the design and materials used in the sweat-based sensors can also play a crucial role in their effectiveness. For example, in cold environments, sensors may need to be insulated to keep them warm and prevent them from freezing. In hot environments, sensors may need to be made with materials that are resistant to moisture and heat, such as breathable fabrics that allow for better air flow and cooling.

7. "Aging of the materials needs to be considered, whereby signals are no longer transmitted efficiently" please explain further. what is the meaning of aging of the material?

Author response: Thank you for the feedback. The paragraph on ageing of materials comprising sensors has been expanded to be clearer on what we imply by ageing. We now also provide more precise examples of ageing of different materials comprising the sensors.

Author Action: We clarified this point by adding the following.

The ideal sensor must be stable and durable to withstand continuous wear and changing environmental conditions such as exposure to water. Ageing, or degradation over time in storage or use, of biological products such as enzymes and antibodies used in biosensors can lead to decreased sensitivity and low reliability¹³⁰. Although biosensor ageing is a well-studied issue, stabilizing biological components in sensors remains a significant challenge for successful implementation in devices¹³¹. Not only biological, but also polymeric and inorganic components of sensors can degrade over time in storage and/or use¹³⁰. Therefore, the ageing of materials must be considered, especially for textile wearable sensors that are subject to more mechanical stress during usage and washing¹³². Signals can no longer be transmitted efficiently when materials degrade, which can impact the reliability of the sensor.

8. "For example, rapid fouling in sweat remains a considerable problem in fine techniques like aptasensing" What are fine techniques? Why aptasensors are part of it? Please explain

Author response: Thank you for your comment! Indeed, the phrase was not clear in the previous version of the manuscript. What we have meant to say was that analytic techniques that deal with analyte-receptor interactions (including aptasensing) on the interface are more impacted by the surface fouling. Other sensor types, are of course, affected by fouling in sweat too, as we discuss in the other section of the review. We now provide a phrase on current developments that aim to solve the fouling problem in sweat sensors using modern materials.

Author Action: We clarified this point by adding the following.

The proposed technologies for use in a future multiplexed sensor are well-established technologies, capable of detecting specific biomarkers in sweat with robustness. However, further research is necessary to translate these foundational technologies into a comprehensive multiplexed sensing device. One challenge is the rapid biofouling that occurs in sweat, particularly in techniques that depend on molecular analyte-receptor interactions at the interface, such as molecularly imprinted polymers and aptasensors¹⁴⁰. Additionally, contamination from sweat and skin can have a significant impact on sensor performance, as discussed in the section on enzymatic sensors. Advanced materials are being developed to address these challenges by incorporating antifouling properties and self-cleaning capabilities¹⁴¹. This ongoing research aims to improve the robustness and reliability of future multiplexed sensors.

9. “crosstalk may become an issue that can be solved through appropriate data acquisition electronics” What about chemical cross-talking? For example, in your proposed sensor in figure 2, there might be chemical crosstalking between glucose and uric acid if the measurement is based on hydrogen peroxide coming from both reactions. Please discuss.

Author response: Thank you for highlighting this valid point. We have now significantly extended this paragraph, providing a description of chemical crosstalk phenomenon and providing examples of how it is avoided in the state-of-the-art literature. While there are several strategies to achieve so, we briefly mention the microfluidic approach that allows one to separate the sweat flow into several channels for each sensor.

Author Action: We clarified this point by adding the following.

Recent reviews of multisensory found that they enhance the overall predictive ability to detect relevant health events, acting in unison, with greater reliability in their readings^{143,144}. However, issues such as comfort also need to be considered as not all patients may be comfortable with wearing multiple sensors that collect different measures¹⁴³. The integration of multiple sensing modalities may result in electronics crosstalk, which can be resolved with appropriate data acquisition electronics. However, chemical crosstalk can still be a potential issue when multiplexing biosensors on the same chip, particularly in the case of enzymatic sensors that generate hydrogen peroxide upon interaction with the analyte. Recent studies have proposed microfluidic systems as a solution to this problem, where sweat flow can be split into separate channels corresponding to each individual sensor to prevent any chemical crosstalk and mixing of samples. This approach allows different sensors to analyze the same sweat sample simultaneously and in real-time. Such a strategy can be used to multiplex a wide range of sensors on a single microfluidic device¹⁴⁵.

10. Figure 3, please discuss further the “cold and hot environment” placements.

Author response: Thank you for pointing this out.

Author Action: We clarified this point by adding the following.

The placement of wearables and sensing textiles for cold environments versus hot environments is of utmost importance in ensuring their efficacy and accuracy in collecting data. For instance, in cold environments, sweat production may decrease, and the sweat-based sensors may not work as effectively. In these situations, it may be necessary to place the sensors in areas where the skin is warmer and more likely to produce sweat, such as the palms of the hands, as shown in Figure 2. Placing the sensors in these areas can help ensure that there is enough sweat for the sensors to analyze.

In hot environments, excessive sweating can cause issues with sweat-based sensors. In these conditions, it is essential to place the sensors in areas that are less prone to sweating, such as the soles of the feet as shown in Figure 2, where they are less likely to be affected by excessive moisture. Additionally, the sensors should be placed in a well-ventilated area to prevent heat buildup, which can cause them to malfunction.

Note that these sensors typically work by analyzing the sweat droplets that accumulate on the skin surface or are collected by the sensor. However, when the body overheats, sweat production can increase significantly,

and this can cause issues for sweat-based sensors. Excessive sweating can lead to a buildup of sweat on the sensor, which can interfere with the accuracy of the readings. Additionally, high temperatures can cause the sensor to malfunction or even stop working altogether. Therefore, it is recommended that wearables and textiles be placed in areas where they are less likely to be exposed to external factors, temperature, humidity, and physical contact, which can affect their accuracy and reliability.

In both hot and cold environments, the design and materials used in the sweat-based sensors can also play a crucial role in their effectiveness. For example, in cold environments, sensors may need to be insulated to keep them warm and prevent them from freezing. In hot environments, sensors may need to be made with materials that are resistant to moisture and heat, such as breathable fabrics that allow for better air flow and cooling.

11. “However, despite recent studies demonstrating that the composition of naturally produced and heat- or pharmaceutically-induced sweat is almost identical, sweat induction via iontophoretic or local heating methods may still be uncomfortable for many device wearers who would prefer passive collection, if possible” Please give examples and discuss passive sweat collection

Author response: Thank you for the great suggestion.

Author Action: We clarified this point by adding the following.

Although passive sweat collection may seem more user-friendly as it doesn't cause any discomfort, there are also some drawbacks to this method. Studies have shown that longer periods of sweat collection are required to obtain the same or smaller amounts of sweat compared to active induction methods. This may not be an issue for stable aqueous analytes, but for metabolites that degrade quickly after excretion, prolonged collection time may lead to measurement inaccuracies. However, as sensors become increasingly miniaturized, smaller amounts of sweat are required for efficient analysis, potentially mitigating the problem of low sweat collection rates in passive mode¹⁵⁹

12. Please consider including sensing technologies that are still not wearable as an opportunity to create the most comprehensive MetS wearable. The current perspective section is just a repetition of the previous discussion. In the future, new technologies should be proposed. For example, there are many enzymatic sensors for measuring cholesterol and triglycerides, there are epidermal sensors able to measure blood pressure, there are commercial devices able to measure the fat under the skin using skin resistance, etc. Figure 2 should be completely modified to include a comprehensive device and a better representation of the sensor. This perspective should guide and inspire future researchers, thus, the future device for MetS should be complete at least in concept. Try including how the signal would be transmitted, and utilized by the wearer or caregiver, including a diagram, etc. The figure should include, all the parameters considered important to be monitored by a MetS wearable device.

Author response: Thank you for the valuable suggestion.

Author Action 1: We added a new figure, Figure 2.

Author Action 2: We clarified this point by adding the following.

Most state-of-the-art developments regarding wearable real-time biomarker detection rely on electrochemical techniques that are robust, have low detection limits, and can be easily miniaturized and integrated into electronic components.¹³⁵ Although biomarkers represent different chemical classes and require different analytical techniques, there is potential for a consolidated sensor. Figure 2 illustrates the full cycle of sweat production and monitoring in a proposed futuristic technology. The figure starts by showing how sweat is produced by sweat glands and travels to the surface of the skin. The proposed technology involves a wearable patch that combines three different technologies to detect five metabolic biomarkers in the sweat. These biomarkers include glucose, lactate, sodium, potassium, and cortisol, which are continuously monitored and processed on the cloud via artificial intelligence. The data is then used to provide personalized feedback and recommendations to the user, such as advice on healthy eating or physical activity. This figure provides a comprehensive overview of the proposed technology and highlights the potential of sweat-based wearable technology for monitoring metabolic health in a non-invasive and personalized way.

Reviewer #5

This review paper shows that metabolic syndrome can be evaluated with wearable sweat sensors. I think this is an important paper that shows the applicability of many previously developed wearable sweat sensors. The author organized metabolic diseases well and classified them according to the substance to analyze wearable sensors. I think this paper is systematically written and will be of great help to readers. If possible, organizing the representative images of wearable sweat sensors, their adhesion to the skin, and how to collect sweat through pictures will help readers understand easily.

Author response: Thank you for the valuable feedback and great suggestion.

Author Action: We removed Table 2, and we added a new figure to show a full cycle from sweat data collection to providing recommendations to the user.

Reviewers' comments:

Reviewer #4 (Remarks to the Author):

The authors satisfactorily addressed the comments. However, they raised some new important questions related to the discussions added, please see below:

1. "Thus, a future integrated sensor for metabolic syndrome would ideally collect multiple data on levels of glucose, sodium, CRP, uric acid, and cortisol." This is not the ideal scenario. The ideal sensor should measure as many parameters as possible. Please justify why only these analytes
2. "...This difference in sweat rates can affect the accuracy of metabolite analysis, with higher sweat rates leading to more accurate results for some low-concentration metabolites..." Please explain. It is expected that a high sweat rate would lead to dilution effects, which is the worst scenario for the low-concentration metabolites.
3. Figure 2. Was it supposed to have a scale for sweat gland density? color scale?
4. Regarding the discussion about the placement of the sensor: please clarify that this is only true for sweat sensors using passive sweat. Otherwise, chemically stimulated sweat should be independent of the weather.
5. "... Additionally, the sensors should be placed in a well-ventilated area to prevent heat buildup, which can cause them to malfunction." why does the device/sensor would malfunction with heat? please clarify. Related to the enzyme activity?
6. "Excessive sweating can lead to a buildup of sweat on the sensor, which can interfere with the accuracy of the readings." The authors stated that a high flow rate is better for sensing low concentration analytes. please explain, seems a contradiction.
7. Why would heat make the sensor "even stop working altogether"?
8. "For example, in cold environments, sensors may need to be insulated to keep them warm and prevent them from freezing." They are in contact with human skin, not likely to freeze. Please revise

Reviewer #4

1. "Thus, a future integrated sensor for metabolic syndrome would ideally collect multiple data on levels of glucose, sodium, CRP, uric acid, and cortisol." This is not the ideal scenario. The ideal sensor should measure as many parameters as possible. Please justify why only these analytes.

Author response: Thank you for your valuable feedback.

Author Action: We clarified this point by modifying the sentence as follows.

"A future integrated sensor for metabolic syndrome should ideally measure as many parameters as possible to obtain a comprehensive assessment of an individual's metabolic health. However, for the purpose of this study, we focused on five specific analytes (glucose, sodium, CRP, uric acid, and cortisol) that have been identified as relevant for the diagnosis and management of metabolic syndrome according to the National Institutes of Health (NIH) definition provided in 2020 (detailed in Table 1). It is worth noting that the optimal sensor for metabolic syndrome may vary depending on the specific diagnostic criteria and patient population being studied."

2. "...This difference in sweat rates can affect the accuracy of metabolite analysis, with higher sweat rates leading to more accurate results for some low-concentration metabolites..." Please explain. It is expected that a high sweat rate would lead to dilution effects, which is the worst scenario for the low-concentration metabolites.

Author response: Thank you for your valuable feedback.

Author Action: We clarified this point by modifying the sentence as follows.

"A high sweat rate could potentially lead to dilution effects, which could negatively impact the accuracy of metabolite analysis for low-concentration metabolites. However, some low-concentration metabolites may not be present enough in the sweat to be accurately detected and quantified, particularly if the sweat rate is too low. In these cases, a higher sweat rate can help to increase the concentration of these metabolites in the sweat, making them easier to detect and quantify. None of the five metabolites (glucose, sodium, CRP, uric acid, and cortisol) are consistently found in high concentrations in sweat. Additionally, the relationship between sweat rate and the concentration of these metabolites in sweat is not well-established. In general, however, it is essential to consider the impact of sweat rate on metabolite analysis carefully and to optimize experimental conditions accordingly."

3. Figure 2. Was it supposed to have a scale for sweat gland density? color scale?

Author response: Thank you for your valuable feedback.

Author Action: A colormap is added to the figure.

4. Regarding the discussion about the placement of the sensor: please clarify that this is only true for sweat sensors using passive sweat. Otherwise, chemically stimulated sweat should be independent of the weather.

Author response: Thank you for your valuable feedback.

Author Action: We clarified this point by modifying the sentence as follows.

“In other words, passive sweat sensors rely on the natural flow of sweat from the skin, and environmental factors such as heat and humidity can affect the amount and flow of sweat produced, which can impact the accuracy of the sensor readings. On the other hand, active sweat sensors, which use chemicals to stimulate the production of sweat, may not be as affected by weather conditions. These types of sensors rely on a chemical reaction to trigger sweat production, and the resulting sweat is often more consistent in volume and flow rate, regardless of external environmental factors. However, it is important to note that the placement of any type of sensor should still be carefully considered to ensure optimal accuracy and reliable measurements. Factors such as the proximity to sweat glands, skin temperature, and skin type can all affect the performance of the sensor, and proper placement can help to minimize these potential sources of error.”

5. “... Additionally, the sensors should be placed in a well-ventilated area to prevent heat buildup, which can cause them to malfunction.” why does the device/sensor would malfunction with heat? please clarify. Related to the enzyme activity?

Author response: Thank you for your valuable feedback.

Author Action: We clarified this point by modifying the sentence as follows.

Heat can cause electronic sensors to malfunction or even stop working altogether for several reasons:

- High temperatures can cause the electrical components inside the sensor to expand, leading to the deformation or breakage of internal structures.
- Heat can cause thermal noise, interfering with the sensor's signal.
- Excessive heat can cause the material used in the sensor's construction to degrade.

Therefore, placing the sensors in a well-ventilated area is essential to prevent heat buildup, which can cause them to malfunction.

6. "Excessive sweating can lead to a buildup of sweat on the sensor, which can interfere with the accuracy of the readings." The authors stated that a high flow rate is better for sensing low concentration analytes. please explain, seems a contradiction.

Author response: Thank you for your valuable feedback.

Author Action: We clarified this point by modifying the sentence as follows.

"Excessive sweating can create a barrier between the sensor and the metabolite being measured, leading to inaccurate readings. A high flow rate of the metabolite can help to increase the chances of the analyte molecules reaching the sensor surface and interacting with it, resulting in a more accurate measurement, especially for low concentration metabolite."

7. Why would heat make the sensor "even stop working altogether"?

Author response: Thank you for your valuable feedback.

Author Action: We addressed in point #5.

8. "For example, in cold environments, sensors may need to be insulated to keep them warm and prevent them from freezing." They are in contact with human skin, not likely to freeze. Please revise

Author response: Thank you for your valuable feedback.

Author Action: We clarified this point by modifying the sentence as follows.

"In cold environments, it may be necessary to provide additional insulation around the sensors to prevent any temperature changes that could affect their accuracy. Sensors in contact with human skin are not likely to freeze in cold environments. However, extreme cold can still affect the sensor's performance by decreasing sensitivity or causing other malfunctions. In addition, if the sensor is located in an anatomical site exposed to cold air, such as fingers or toes, it may be more susceptible to cooling and require insulation to maintain its temperature and performance. Overall, while the risk of freezing may be low, it is still essential to consider the effects of cold temperatures on sensor performance and take appropriate measures to ensure accurate measurements."